## PROCEEDINGS A

applied mathematics, mathematical modelling, computational physics

dynamic mode decomposition, time-delay coordinates, Frenet–Serret, Koopman operator, Hankel matrix

**Author for correspondence:**
Seth M. Hirsh
e-mail: hirshs@uw.edu

# Structured time-delay models for dynamical systems with connections to Frenet–Serret frame

Seth M. Hirsh[1], Sara M. Ichinaga[2], Steven L. Brunton[3], J. Nathan Kutz[2] and Bingni W. Brunton[4]

[1]Department of Physics, [2]Department of Applied Mathematics, [3]Department of Mechanical Engineering, and [4]Department of Biology, University of Washington, Seattle, WA USA

SMH, 0000-0002-6892-9508; SLB, 0000-0002-6565-5118; JNK, 0000-0002-6004-2275; BWB, 0000-0002-4831-3466

Time-delay embedding and dimensionality reduction are powerful techniques for discovering effective coordinate systems to represent the dynamics of physical systems. Recently, it has been shown that models identified by dynamic mode decomposition on time-delay coordinates provide linear representations of strongly nonlinear systems, in the so-called Hankel alternative view of Koopman (HAVOK) approach. Curiously, the resulting linear model has a matrix representation that is approximately antisymmetric and tridiagonal; for chaotic systems, there is an additional forcing term in the last component. In this paper, we establish a new theoretical connection between HAVOK and the Frenet–Serret frame from differential geometry, and also develop an improved algorithm to identify more stable and accurate models from less data. In particular, we show that the sub- and super-diagonal entries of the linear model correspond to the intrinsic curvatures in the Frenet–Serret frame. Based on this connection, we modify the algorithm to promote this antisymmetric structure, even in the noisy, low-data limit. We demonstrate this improved modelling procedure on data from several nonlinear synthetic and real-world examples.

# 1. Introduction

Discovering meaningful models of complex, nonlinear systems from measurement data has the potential to improve characterization, prediction and control. Focus has increasingly turned from first-principles modelling towards data-driven techniques to discover governing equations that are as simple as possible while accurately describing the data [1–4]. However, available measurements may not be in the right coordinates for which the system admits a simple representation. Thus, considerable effort has gone into learning effective coordinate transformations of the measurement data [5–7], especially those that allow nonlinear dynamics to be approximated by a linear system. These coordinates are related to eigenfunctions of the Koopman operator [8–13], with dynamic mode decomposition (DMD) [14] being the leading computational algorithm for high-dimensional spatio-temporal data [11,13,15]. For low-dimensional data, time-delay embedding [16] has been shown to provide accurate linear models of nonlinear systems [5,17,18]. Linear time-delay models have a rich history [19,20], and recently, DMD on delay coordinates [15,21] has been rigorously connected to these linearizing coordinate systems in the Hankel alternative view of Koopman (HAVOK) approach [5,7,17]. In this work, we establish a new connection between HAVOK and the Frenet–Serret frame from differential geometry, which inspires an extension to the algorithm that improves the stability of these models.

Time-delay embedding is a widely used technique to characterize dynamical systems from limited measurements. In delay embedding, incomplete measurements are used to reconstruct a representation of the latent high-dimensional system by augmenting the present measurement with a time history of previous measurements. Takens showed that under certain conditions, time-delay embedding produces an attractor that is diffeomorphic to the attractor of the latent system [16]. Time-delay embeddings have also been extensively used for signal processing and modelling [19,20,22–27], for example, in singular spectrum analysis (SSA) [19,22] and the eigensystem realization algorithm (ERA) [20]. In both cases, a time history of augmented delay vectors are arranged as columns of a Hankel matrix, and the singular value decomposition (SVD) is used to extract *eigen*-time-delay coordinates in a dimensionality reduction stage. More recently, these historical approaches have been connected to the modern DMD algorithm [15], and it has become commonplace to compute DMD models on time-delay coordinates [15,21]. The HAVOK approach established a rigorous connection between DMD on delay coordinates and eigenfunctions of the Koopman operator [5]; HAVOK [5] is also referred to as Hankel DMD [17] or delay DMD [15].

HAVOK produces linear models where the matrix representation of the dynamics has a peculiar and particular structure. These matrices tend to be skew-symmetric and dominantly tridiagonal, with zero diagonal (see figure 1 for an example). In the original HAVOK paper, this structure was observed in some systems, but not others, with the structure being more pronounced in noise-free examples with an abundance of data. It has been unclear how to interpret this structure and whether or not it is a universal feature of HAVOK models. Moreover, the eigen-time-delay modes closely resemble Legendre polynomials; these polynomials were explored further in Kamb *et al.* [28]. The present work directly resolves this mysterious structure by establishing a connection to the Frenet–Serret frame from differential geometry.

The structure of HAVOK models may be understood by introducing intrinsic coordinates from differential geometry [29]. One popular set of intrinsic coordinates is the Frenet–Serret frame, which is formed by applying the Gram–Schmidt procedure to the derivatives of the trajectory $\dot{x}(t), \ddot{x}(t), \dddot{x}(t), \ldots$ [30–32]. Álvarez-Vizoso *et al.* [33] showed that the SVD of trajectory data converges locally to the Frenet–Serret frame in the limit of an infinitesimal time step. The Frenet–Serret frame results in an orthogonal basis of polynomials, which we will connect to the observed Legendre basis of HAVOK [5,28]. Moreover, we show that the dynamics, when represented in these coordinates, have the same tridiagonal structure as the HAVOK models. Importantly, the terms along the sub- and super-diagonals have a specific physical interpretation as intrinsic curvatures. By enforcing this structure, HAVOK models are more robust to noisy and limited data.

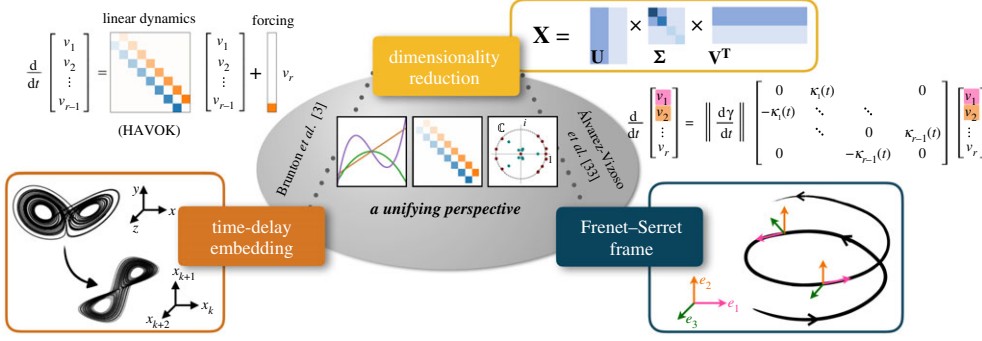

**Figure 1.** In this work, we unify key results from dimensionality reduction, time-delay embedding and the Frenet–Serret frame to show that a dynamical system may be decomposed into a sparse linear model plus a forcing term. Furthermore, this linear model has a particular structure: it is an antisymmetric matrix with non-zero elements only along the super- and sub-diagonals. These non-zero elements are interpretable as they are intrinsic curvatures of the system in the Frenet–Serret frame.

In this work, we present a new theoretical connection between time-delay embedding models and the Frenet–Serret frame from differential geometry. Our unifying perspective sheds light on the antisymmetric, tridiagonal structure of the HAVOK model. We use this understanding to develop *structured* HAVOK models that are more accurate for noisy and limited data. Section 2 provides a review of dimensionality reduction methods, time-delay embeddings and the Frenet–Serret frame. This section also discusses current connections between these fields. In §3, we establish the main result of this work, connecting linear time-delay models with the Frenet–Serret frame, explaining the tridiagonal, antisymmetric structure seen in figure 1. We then illustrate this theory on a synthetic example. In §4, we explore the limitations and requirements of the theory, giving recommendations for achieving this structure in practice. In §5, based on this theory, we develop a modified HAVOK method, called *structured* HAVOK (sHAVOK), which promotes tridiagonal, antisymmetric models. We demonstrate this approach on three nonlinear synthetic examples and two real-world datasets, namely measurements of a double pendulum experiment and measles outbreak data, and show that sHAVOK yields more stable and accurate models from significantly less data.

## 2. Related work

Our work relates and extends results from three fields: dimensionality reduction, time-delay embedding and the Frenet–Serret coordinate frame from differential geometry. There is an extensive literature on each of these fields, and here we give a brief introduction of the related work to establish a common notation on which we build a unifying framework in §3.

## (a) Dimensionality reduction

Recent advancements in sensor and measurement technologies have led to a significant increase in the collection of time-series data from complex, spatio-temporal systems. Although such data are typically high dimensional, in many cases, it can be well approximated with a low-dimensional representation. One central goal is to learn the underlying structure of this data. Although there are many data-driven dimensionality reduction methods, here we focus on linear techniques because of their effectiveness and analytic tractability. In particular, given a data matrix $X \in \mathbb{R}^{m \times n}$, the goal of these techniques is to decompose $X$ into the matrix product

$$X = UV^{\mathsf{T}}, \tag{2.1}$$

where $U \in \mathbb{R}^{m \times k}$ and $V \in \mathbb{R}^{n \times k}$ are low rank ($k < \min(m, n)$). The task of solving for $U$ and $V$ is highly underdetermined, and different solutions may be obtained when different assumptions are made.

Here, we review two popular linear dimensionality reduction techniques: SVD [34,35] and DMD [13,15,36]. Both of these methods are key components of the HAVOK algorithm and play a key role in determining the underlying tridiagonal antisymmetric structure in figure 1.

### (i) SVD

The SVD is one of the most popular dimensionality reduction methods, and it has been applied in a wide range of applications, including genomics [37], physics [38] and image processing [39]. SVD is the underlying algorithm for *principal component analysis*.

Given the data matrix $X \in \mathbb{R}^{m \times n}$, the SVD decomposes $X$ into the product of three matrices,

$$X = U \Sigma V^{\mathsf{T}},$$

where $U \in \mathbb{R}^{m \times m}$ and $V \in \mathbb{R}^{n \times n}$ are unitary matrices, and $\Sigma \in \mathbb{R}^{m \times n}$ is a diagonal matrix with non-negative entries [34,35]. We denote the $i$th columns of $U$ and $V$ as $u_i$ and $v_i$, respectively. The diagonal elements of $\Sigma$, $\sigma_i$, are known as the singular values of $X$, and they are written in descending order.

The rank of the data is defined to be $R$, which equals the number of non-zero singular values. Consider the low-rank matrix approximation

$$X_r = \sum_{j=1}^{r} u_j \sigma_j v_j^T,$$

with $r \leq R$. An important property of $X_r$ is that it is the best rank $r$ approximation to $X$ in the least-squares sense. In other words,

$$X_r = \mathrm{argmin}_Y \|X - Y\| \quad \text{such that rank}(Y) = r,$$

with respect to both the $l_2$ and Frobenius norms. Furthermore, the relative error in this rank-$r$ approximation using the $l_2$ norm is

$$\frac{\|X - X_r\|_{l_2}}{\|X\|_{l_2}} = \frac{\sigma_{r+1}}{\sigma_1}. \tag{2.2}$$

From (2.2), we immediately see that if the singular values decay rapidly ($\sigma_{j+1} \ll \sigma_j$), then $X_r$ is a good low-rank approximation to $X$. This property makes the SVD a popular tool for compressing data.

### (ii) DMD

DMD [13–15] is another linear dimensionality reduction technique that incorporates an assumption that the measurements are time-series data generated by a linear dynamical system in time. DMD has become a popular tool for modelling dynamical systems in such diverse fields, including fluid mechanics [11,14], neuroscience [21], disease modelling [40], robotics [41], plasma modelling [42], resolvent analysis [43] and computer vision [44,45].

Like the SVD, for DMD, we begin with a data matrix $X \in \mathbb{R}^{m \times n}$. Here, we assume that our data are generated by an unknown dynamical system so that the columns of $X$, $x(t_k)$, are time snapshots related by the map $x(t_{k+1}) = F(x(t_k))$. While $F$ may be nonlinear, the goal of DMD is to determine the best-fit linear operator $A : \mathbb{R}^m \to \mathbb{R}^m$ such that

$$x(t_{k+1}) \approx A x(t_k).$$

If we define the two time-shifted data matrices,

$$X_1^{n-1} = \begin{bmatrix} | & | & \cdots & | \\ x(t_1) & x_2(t_2) & \cdots & x(t_{n-1}) \\ | & | & \cdots & | \end{bmatrix} \quad \text{and} \quad X_2^n = \begin{bmatrix} | & | & \cdots & | \\ x(t_2) & x(t_3) & \cdots & x(t_n) \\ | & | & \cdots & | \end{bmatrix},$$

then we can equivalently define $A \in \mathbb{R}^{m \times m}$ to be the operator such that

$$X_2^n \approx A X_1^{n-1}.$$

It follows that $A$ is the solution to the minimization problem

$$A = \min_{A'} \| X_2^n - A' X_1^{n-1} \|_F,$$

where $\|\cdot\|_F$ denotes the Frobenius norm.

A unique solution to this problem can be obtained using the *exact DMD* method and the Moore–Penrose pseudo-inverse $\hat{A} = X_2^n (X_1^{n-1})^\dagger$ [13,15]. Alternative algorithms have been shown to perform better for noisy measurement data, including optimized DMD [46], forward–backward DMD [47] and total least-squares DMD [48].

One key benefit of DMD is that it builds an explicit temporal model and supports short-term future state prediction. Defining $\{\lambda_j\}$ and $\{v_j\}$ to be the eigenvalues and eigenvectors of $A$, respectively, then we can write

$$x(t_k) = \sum_{j=1}^{r} v_j \, e^{\omega_j t_k}, \tag{2.3}$$

where $\omega_j = \ln(\lambda_j)/\Delta t$ are eigenvalues normalized by the sampling interval $\Delta t$, and the eigenvectors are normalized such that $\sum_{j=1}^{r} v_j = x(t_1)$. Thus, to compute the state at an arbitrary time $t$, we can simply evaluate (2.3) at that time. Furthermore, letting $v_j$ be the columns of $U$ and $\{\exp(\omega_j t_k) \text{ for } k = 1, \dots r\}$ be the columns of $V$, then we can express data in the form of (2.1).

## (b) Time-delay embedding

Suppose we are interested in a dynamical system

$$\frac{d\xi}{dt} = F(\xi),$$

where $\xi(t) \in \mathbb{R}^l$ are states whose dynamics are governed by some unknown nonlinear differential equation. Typically, we measure some possibly nonlinear projection of $\xi$, $x(\xi) \in \mathbb{R}^d$ at discrete time points $t = 0, \Delta t, \dots, q\Delta t$. In general, the dimensionality of the underlying dynamics is unknown, and the choice of measurements are limited by practical constraints. Consequently, it is difficult to know whether the measurements $x$ are sufficient for modelling the system. For example, $d$ may be smaller than $m$. In this work, we are primarily interested in the case of $d = 1$; in other words, we have only a single one-dimensional time-series measurement for the system.

We can construct an embedding of our system using successive time delays of the measurement $x$, at $x(t - \tau)$. Given a single measurement of our dynamical system $x(t) \in \mathbb{R}$, for $t = 0, \Delta t, \dots (q - 1)\Delta t$, we can form the Hankel matrix $H \in \mathbb{R}^{m \times n}$ by stacking time-shifted snapshots of $x$ [49],

$$H = \begin{bmatrix} x_1 & x_2 & x_3 & x_4 & \cdots & x_n \\ x_2 & x_3 & x_4 & x_5 & \cdots & x_{n+1} \\ \vdots & \vdots & \vdots & \vdots & \ddots & \vdots \\ x_m & x_{m+1} & x_{m+2} & x_{m+3} & \cdots & x_q \end{bmatrix}. \tag{2.4}$$

Each column may be thought of as an augmented state space that includes a short, $m$-dimensional trajectory in time. Our data matrix $H$ is then this $m$-dimensional trajectory measured over $n$ snapshots in time.

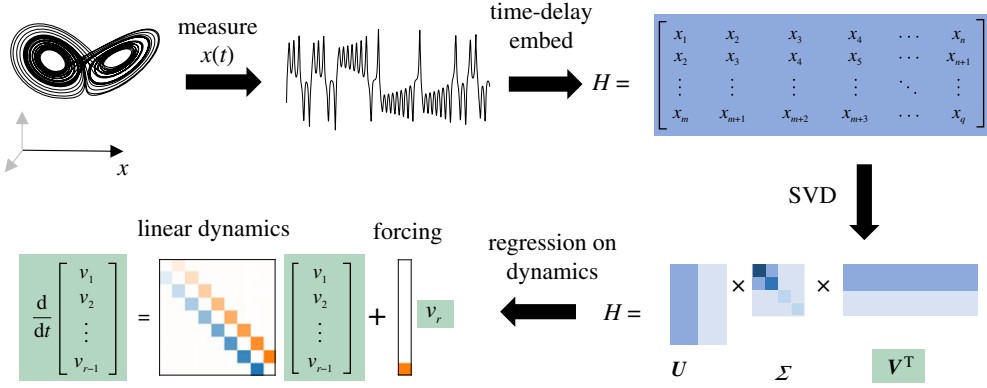

**Figure 2.** Outline of steps in HAVOK method. First, given a dynamical system a single variable $x(t)$ is measured. Time-shifted copies of $x(t)$ are stacked to form a Hankel matrix $\boldsymbol{H}$. The singular value decomposition (SVD) is applied to $\boldsymbol{H}$, producing a low-dimensional representation $\boldsymbol{V}$. The dynamic mode decomposition (DMD) is then applied to $\boldsymbol{V}$ to form a linear dynamical model and a forcing term.

There are several key benefits of using time-delay embeddings. Most notably, given a chaotic attractor, Taken's embedding theorem states that a sufficiently high-dimensional time-delay embedding of the system is diffeomorphic to the original attractor [16], as illustrated in figure 2. In addition, recent results have shown that time-delay matrices are guaranteed to have strongly decaying singular value spectra. In particular, Beckerman & Townsend [50] prove the following theorem:

**Theorem 2.1.** *Let $\boldsymbol{H}_n \in \mathbb{R}^{n \times n}$ be a positive definite Hankel matrix, with singular values $\sigma_1, \ldots, \sigma_n$. Then $\sigma_j \leq C \rho^{-j/\log n} \sigma_1$ for constants $C$ and $\rho$ and for $j = 1, \ldots, n$.*

Equivalently, $\boldsymbol{H}_n$ can be approximated up to an accuracy of $\epsilon \|\boldsymbol{H}_n\|_2$ by a rank $\mathcal{O}(\log n \log 1/\epsilon)$ matrix. From this, we see that $\boldsymbol{H}_n$ can be well-approximated by a low-rank matrix.

Many methods have been developed to take advantage of this structure of the Hankel matrix, including the *ERA* [20], SSA [19] and nonlinear Laplacian spectrum analysis [22]. DMD may also be computed on delay coordinates from the Hankel matrix [15,21,51], and it has been shown that this approach may provide a Koopman invariant subspace [5,52]. In addition, this structure has also been incorporated into neural network architectures [53].

This analysis is limited to delay embeddings of one-dimensional signals. However, embeddings of multi-dimensional signals have also been explored [15,54]. Most notably, *higher order DMD* is particularly powerful for very high dimensional embeddings [55–57]. Understanding the structure of these higher dimensional embeddings is also an exciting area of current research.

## (c) HAVOK: dimensionality reduction and time-delay embeddings

Leveraging dimensionality reduction and time-delay embeddings, the HAVOK algorithm constructs low-dimensional models of dynamical systems [5]. Specifically, HAVOK learns effective measurement coordinates of the system and estimates its intrinsic dimensionality. Remarkably, HAVOK models are simple, consisting of a linear model and a forcing term that can be used for short-term forecasting.

We illustrate this method in figure 1 for the Lorenz system (see §5b for details about this system). To do so, we begin with a one-dimensional time series $x(t)$ for $t = 0, \Delta t, \ldots, (q-1)\Delta t$. We construct a higher dimensional representation using time-delay embeddings, producing a

Hankel matrix $H \in \mathbb{R}^{m \times n}$ as in (2.4) and compute its SVD,

$$H = U \Sigma V^{\mathsf{T}}.$$

If $H$ is sufficiently low rank (with rank $r$), then we need only consider the reduced SVD,

$$H_r = U_r \Sigma_r V_r^{\mathsf{T}},$$

where $U_r \in \mathbb{R}^{m \times r}$ and $V_r \in \mathbb{R}^{n \times r}$ are orthogonal matrices and $\Sigma_r \in \mathbb{R}^{r \times r}$ is diagonal. Rearranging the terms, $V_r^{\mathsf{T}} = \Sigma_r^{-1} U_r^{\mathsf{T}} H_r$ and we can think of

$$V_r^{\mathsf{T}} = \begin{bmatrix} v_1 & v_2 & \cdots & v_n \end{bmatrix} \tag{2.5}$$

as a lower dimensional representation of our high dimensional trajectory. For quasi-periodic systems, the SVD decomposition of the Hankel matrix results in *principal component trajectories* [54], which reconstruct dynamical trajectories in terms of periodic orbits.

To discover the linear dynamics, we apply DMD. In particular, we construct the time-shifted matrices,

$$V_1^{\mathsf{T}} = \begin{bmatrix} v_1 & v_2 & \cdots & v_{n-1} \end{bmatrix} \quad \text{and} \quad V_2^{\mathsf{T}} = \begin{bmatrix} v_2 & v_3 & \cdots & v_n \end{bmatrix}. \tag{2.6}$$

We then compute the linear approximation $\hat{A}$ such that $V_2^{\mathsf{T}} = \hat{A} V_1^{\mathsf{T}}$, where $\hat{A} = V_2^{\mathsf{T}} V_1^{\mathsf{T}^\dagger}$. This yields a model $v_{i+1} = \hat{A} v_i$.

In the continuous case,

$$\dot{v}(t) = A v(t), \tag{2.7}$$

which is related to first order in $\Delta t$ to the discrete case by

$$A \approx \frac{(\hat{A} - I)}{\Delta t}.$$

For a general nonlinear dynamical system, this linear model yields a high RMSE error on the training data. Instead, [5] proposed a linear model plus a forcing term in the last component of $v$ (figure 1):

$$\dot{v}(t) = A v(t) + B v_r(t), \tag{2.8}$$

where $v(t) \in \mathbb{R}^{r-1}$, $A \in \mathbb{R}^{r-1 \times r-1}$ and $B \in \mathbb{R}^{r-1}$. In this case, $V_2$ is defined as columns 2 to $n$ of the SVD singular vectors with an $r-1$ rank truncation $V_{r-1}$. $\hat{A} \in \mathbb{R}^{r-1 \times r-1}$ and $\hat{B} \in \mathbb{R}^{r-1 \times 1}$ are computed as $[\hat{A}, \hat{B}] = V_2^{\mathsf{T}} V_1^{\mathsf{T}^\dagger}$. The continuous analogue of $\hat{B}$, $B$, is computed by $B \approx (\hat{B} - I)/\Delta t$. $v(t)$ corresponds to the first $r-1$ rows of $V_r^{\mathsf{T}}$, while $v_r(t)$ corresponds to the $r$th row of $V_r^{\mathsf{T}}$. The forcing term $v_r$ is required to capture the essential nonlinearity of the system, such as lobe switching, that cannot be captured by the linear model.

Once the $A$ and $B$ matrices have been derived, [5] found that HAVOK models could be used to forecast in an online setting. In particular, given the previous snapshots $x_n, x_{n+1}, \ldots x_q$, we can estimate $v_r$ at the next snapshot by taking the inner product of $x_n, x_{n+1}, \ldots x_q$ with the $r$th column of $U$ scaled by the inverse of the $r$th component of $\Sigma$.

HAVOK was shown to be a successful model for a variety of systems, including a double pendulum, switchings of Earth's magnetic field and measurements of human behaviour [5,58]. In addition, the linear portion of the HAVOK model has been observed to adopt a very particular structure: the dynamics matrix was antisymmetric, with non-zero elements only on the super-diagonal and sub-diagonal (figure 1).

Much work has been done to study the properties of HAVOK. Arbabi *et al.* [17] showed that, in the limit of an infinite number of time delays ($m \to \infty$), $A$ converges to the Koopman operator for ergodic systems. Bozzo *et al.* [59] showed that in a similar limit, for periodic data, HAVOK converges to the temporal discrete Fourier transform. Kamb *et al.* [28] connect HAVOK to the use of convolutional coordinates. The primary goal of this current work is to connect HAVOK to the concept of curvature in differential geometry, and with these new insights, improve the HAVOK algorithm to take advantage of this structure in the dynamics matrix. In contrast to much of the previous work, we focus on the limit where only small amounts of noisy data are available.

## (d) The Frenet–Serret coordinate frame

Suppose we have a smooth curve $\boldsymbol{\gamma}(t) \in \mathbb{R}^m$ measured over some time interval $t \in [a, b]$. As before, we would like to determine an effective set of coordinates in which to represent our data. When using SVD or DMD, the basis discovered corresponds to the spatial modes of the data and is constant in time. However, for many systems, it is sometimes natural to express both the coordinates and basis as functions of time [60,61]. One popular method for developing this non-inertial frame is the Frenet–Serret coordinate system, which has been applied in a wide range of fields, including robotics [62,63], aerodynamics [64] and general relativity [65,66].

Let us assume that $\boldsymbol{\gamma}(t)$ has $r$ non-zero continuous derivatives, $\boldsymbol{\gamma}'(t), \boldsymbol{\gamma}''(t), \ldots \boldsymbol{\gamma}^{(r)}(t)$. We further assume that these derivatives are linearly independent and $\|\boldsymbol{\gamma}'(t)\| \neq \boldsymbol{0}$ for all $t$. Using the Gram–Schmidt process, we can form the orthonormal basis, $e_1, e_2, \ldots, e_r$,

$$
\left.
\begin{aligned}
e_1(t) &= \frac{\boldsymbol{\gamma}'(t)}{\|\boldsymbol{\gamma}'(t)\|}, \\
e_2(t) &= \frac{\boldsymbol{\gamma}''(t) - \langle \boldsymbol{\gamma}''(t), e_1(t) \rangle e_1(t)}{\|\boldsymbol{\gamma}''(t) - \langle \boldsymbol{\gamma}''(t), e_1(t) \rangle e_1(t)\|}, \\
&\vdots \\
\text{and} \quad e_r(t) &= \frac{\boldsymbol{\gamma}^{(r)}(t) - \sum_{k=1}^{r-1} \langle \boldsymbol{\gamma}^{(r)}(t), e_k(t) \rangle e_k(t)}{\left\| \boldsymbol{\gamma}^{(r)}(t) - \sum_{k=1}^{r-1} \langle \boldsymbol{\gamma}^{(r)}(t), e_k(t) \rangle e_k(t) \right\|}.
\end{aligned}
\right\}
\tag{2.9}
$$

Here, $\langle \cdot, \cdot \rangle$ denotes an inner product, and we choose $r \leq m$ so that these vectors are linearly independent and hence form an orthonormal basis. This set of basis vectors define the *Frenet–Serret frame*.

To derive the evolution of this basis, let us define the matrix formed by stacking these vectors $Q(t) = [e_1(t), e_2(t), \ldots, e_r(t)]^\mathsf{T} \in \mathbb{R}^{r \times m}$, so that $Q(t)$ satisfies the following time-varying linear dynamics,

$$
\frac{dQ}{dt} = \|\boldsymbol{\gamma}'(t)\| K(t) Q,
\tag{2.10}
$$

where $K(t) \in \mathbb{R}^{r \times r}$.

By factoring out the term $\|\boldsymbol{\gamma}'(t)\|$ from $K(t)$, it is guaranteed that $K(t)$ does not depend on the parametrization of the curve (i.e. the speed of the trajectory), but only on its geometry. The matrix $K(t)$ is highly structured and sparse. To understand the structure of $K(t)$ we derive two key properties [33]:

(1) $K_{i,j}(t) = -K_{j,i}(t)$ (antisymmetry):

> *Proof.* Since $r \leq m$, then by construction the columns of $Q(t)$ are orthogonal and thus $QQ^\mathsf{T} = I$. Taking the derivative with respect to $t$, $dQ/dt\, Q^T + Q(dQ^\mathsf{T}/dt) = 0$, or equivalently
>
> $$
> \frac{dQ}{dt} Q^\mathsf{T} = -\left( \frac{dQ}{dt} Q^\mathsf{T} \right)^\mathsf{T}.
> $$
>
> Since $Q$ is unitary, then $Q^{-1} = Q^\mathsf{T}$, and hence
>
> $$
> K(t) = \frac{1}{\|\boldsymbol{\gamma}'(t)\|} \frac{dQ}{dt} Q^\mathsf{T},
> $$
>
> from which we immediately see that $K(t) = -K(t)^\mathsf{T}$. ∎

(2) $K_{i,j}(t) = 0$ for $j \geq i + 2$:

We first note that since $e_i(t) \in \text{span}\{\gamma'(t), \ldots, \gamma^i(t)\}$, its derivative must satisfy $e_i'(t) \in \text{span}\{\gamma'(t), \ldots, \gamma^{(i+1)}(t)\}$. Now by construction, using the Gram–Schmidt method, $e_j$ is orthogonal to $\text{span}\{\gamma'(t), \ldots, \gamma^{(i+1)}(t)\}$ for $j \geq i + 2$. Since $e_i'(t)$ is in the span of this set, then $e_j$ must be orthogonal to $e_i'$ for $j \geq i + 2$. Thus, $K_{i,j}(t) = \langle e_i'(t), e_j \rangle = 0$ for $j \geq i + 2$.

With these two constraints, $K(t)$ takes the form,

$$K(t) = \begin{bmatrix} 0 & \kappa_1(t) & & 0 \\ -\kappa_1(t) & \ddots & \ddots & \\ & \ddots & 0 & \kappa_{r-1}(t) \\ 0 & & -\kappa_{r-1}(t) & 0 \end{bmatrix}. \tag{2.11}$$

Thus $K(t)$ is antisymmetric with non-zero elements only along the super-diagonal and sub-diagonal, and the values $\kappa_1(t), \ldots, \kappa_{r-1}(t)$ are defined to be the *curvatures* of the trajectory. The curvatures $\kappa_i(t)$ combined with the basis vectors $e_i(t)$ define the Frenet–Serret apparatus, which fully characterizes the trajectory up to translation [33].

From a geometric perspective, $e_1(t), \ldots, e_r(t)$ form an instantaneous (local) coordinate frame, which moves with the trajectory. The curvatures define how quickly this frame changes with time. If the trajectory is a straight line the curvatures are all zero. If $\kappa_1$ is constant and non-zero, while all other curvatures are zero, then the trajectory lies on a circle. If $\kappa_1$ and $\kappa_2$ are constant and non-zero with all other curvatures zero, then the trajectory lies on a helix. Comparing the structure of (2.11) to figure 1, we immediately see a similarity. Over the following sections, we will shed light on this connection.

## (e) SVD and curvature

Given time-series data, the SVD constructs an orthonormal basis that is fixed in time, whereas the Frenet–Serret frame constructs an orthonormal basis that moves with the trajectory. In recent work, Álvarez-Vizoso *et al.* [33] showed how these frames are related. In particular, the Frenet–Serret frame converges to the SVD frame in the limit as the time interval of the trajectory goes to zero.

To understand this further, consider a trajectory $\gamma(t) \in \mathbb{R}^m$ as described in §2d. If we assume that our measurements are from a small neighbourhood $t \in (-\epsilon, \epsilon)$ (where $\epsilon \ll 1$), then $\gamma(t)$ is well-approximated by its Taylor expansion,

$$\gamma(t) - \gamma(0) = \gamma'(0)t + \frac{\gamma''(0)}{2}t^2 + \frac{\gamma'''(0)}{6}t^3 + \cdots.$$

Writing this in matrix form, we have that

$$\gamma(t) - \gamma(0) = \underbrace{\begin{bmatrix} | & | & | & | \\ \gamma'(0) & \gamma''(0) & \gamma'''(0) & \cdots \\ | & | & | & | \end{bmatrix}}_{\Gamma} \underbrace{\begin{bmatrix} 1 & & & \\ & \frac{1}{2} & & \\ & & \frac{1}{6} & \\ & & & \ddots \end{bmatrix}}_{\Sigma} \underbrace{\begin{bmatrix} - & t & - \\ - & t^2 & - \\ - & t^3 & - \\ - & \vdots & - \end{bmatrix}}_{T^{\mathsf{T}}}. \tag{2.12}$$

Recall one key property of the SVD is that the $r$th rank truncation in the expansion is the best rank-$r$ approximation to the data in the least-squares sense. Since $\epsilon \ll 1$, then each subsequent

term in this expansion is much smaller than the previous term,

$$\|\boldsymbol{\gamma}'(0)t\|_2 \ll \left\|\frac{\boldsymbol{\gamma}''(0)}{2}t^2\right\|_2 \ll \left\|\frac{\boldsymbol{\gamma}'''(0)}{6}t^3\right\|_2 \ll \dots . \tag{2.13}$$

From this, we see that the expansion in (2.12) is strongly related to the SVD. However, in the SVD, we have the constraint that the $U$ and $V$ matrices are orthogonal, while for the Taylor expansion $\boldsymbol{\Gamma}$ and $T$ have no such constraint. Álvarez-Vizoso *et al.* [33] show that in the limit as $\epsilon \to 0$, then $U$ is the result of applying the Gram–Schmidt process to the columns of $\boldsymbol{\Gamma}$, and $V$ is the result of applying the Gram–Schmidt process to the columns of $T$. Comparing this to above, we see that

$$U = \begin{bmatrix} | & | & | & | \\ e_1(0) & e_2(0) & e_3(0) & \cdots \\ | & | & | & | \end{bmatrix} \quad \text{and} \quad V = \begin{bmatrix} | & | & | & | \\ p_1(t) & p_2(t) & p_3(t) & \cdots \\ | & | & | & | \end{bmatrix},$$

where $e_1(t), e_2(t), \dots, e_r(t)$ is the basis for the Frenet–Serret frame defined in (2.9) and

$$p_i(t) = \frac{t^i - \sum_{j=1}^{i-1}\langle t^i, p_j(t)\rangle p_j(t)}{\left\| t^i - \sum_{j=1}^{i-1}\langle t^i, p_j(t)\rangle p_j(t)\right\|} \quad \text{for } i = 1, 2, 3, \dots \tag{2.14}$$

We note that the $p_i(t)$'s form a set of orthogonal polynomials independent of the dataset. In this limit, the curvatures depend solely on the singular values,

$$\kappa_i(t) = \sqrt{a_i}\,\frac{\sigma_{i+1}}{\sigma_1(t)\sigma_i(t)}, \quad \text{where } a_{i-1} = \left(\frac{i}{i+(-1)^i}\right)^2 \frac{4i^2-1}{3}.$$

We note that connections between the SVD and the Gram–Schmidt method are well described in the literature and underlie several different DMD frameworks [15,67]. Furthermore, this particular connection is crucial for understanding the structure in HAVOK models.

# 3. Unifying SVD, time-delay embeddings and the Frenet–Serret frame

In this section, we show that time-series data from a dynamical system may be decomposed into a sparse linear dynamical model with nonlinear forcing, and the non-zero elements along the sub- and super-diagonals of the linear part of this model have a clear geometric meaning: they are curvatures of the system. In §3a, we combine key results about the Frenet–Serret frame, time delays and SVD to explain this structure. Following this theory, §3b illustrates this approach with a simple synthetic example. The decomposition yields a set of orthogonal polynomials that form a coordinate basis for the time-delay embedding. In §3c, we explicitly describe these polynomials and compare their properties with the Legendre polynomials.

## (a) Connecting SVD, time-delay embeddings and Frenet–Serret frame

Here, we connect the properties of the SVD, time-delay embeddings and the Frenet–Serret frame to decompose a dynamical model into a linear dynamical model with nonlinear forcing, where the linear model is both antisymmetric and tridiagonal. To do this, we follow the steps of the HAVOK method with slight modifications and show how they give rise to these structured dynamics. This process is illustrated in figure 3. We emphasize that to develop this new perspective, our key insight is based on deriving a connection between the *global* Koopman frame and the *local* Frenet–Serret frame for the case of time-delay coordinates. To do this, we observe that for a low-dimensional time-delay embedding $H$ that satisfies global analyses, the transpose of this data $H^\mathsf{T}$ is a time-delay embedding. By construction, $H^\mathsf{T}$ covers a short time interval and hence satisfies local analyses. The dynamics of these two sets of data are highly related, since $H$ and $H^\mathsf{T}$ only differ by a transpose, from which we can connect the local/global dynamics. These two perspectives for the same dataset are only possible because we are using time-delay embeddings/Hankel matrices, and the transpose of a Hankel matrix is also a Hankel matrix.

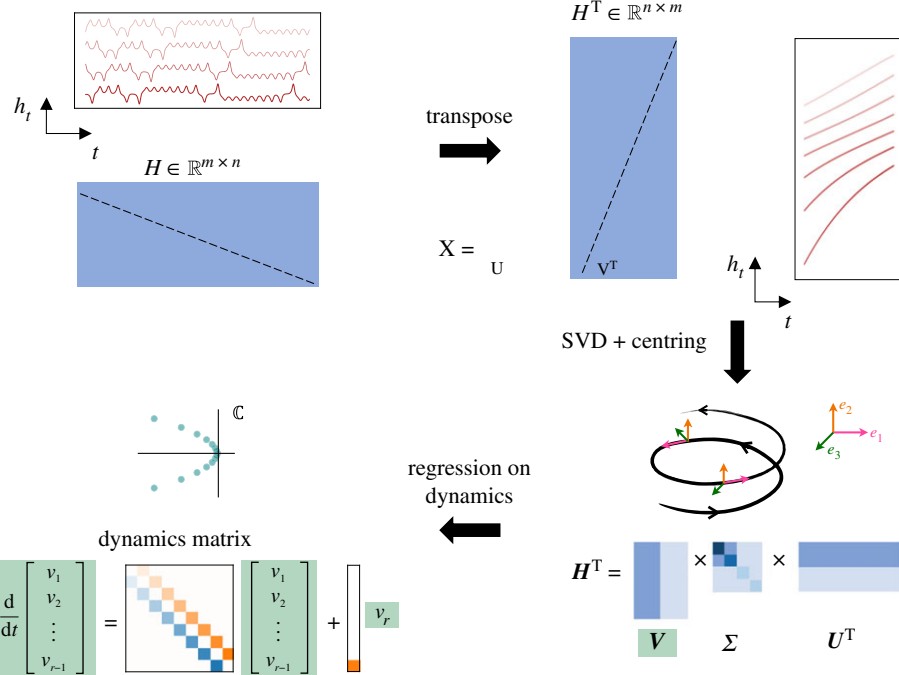

**Figure 3.** An illustration of how a highly structured, antisymmetric linear model arises from time-delay data. Starting with a one-dimensional time series, we construct a $m \times n$ Hankel matrix using time-shifted copies of the data. Assume that $n \gg m$, in which case **H** can be thought of as an $m$ dimensional trajectory over a long period ($n$ snapshots in time). Similarly, the transpose of **H** may be thought of as a high dimensional ($n$ dimensional) trajectory over a short period ($m$ snapshots) in time. With this interpretation, by the results of [33], the singular vectors of **H** after applying centring yield the Frenet–Serret frame. Regression on the dynamics in the Frenet–Serret frame yields the tridiagonal antisymmetric linear model with an additional forcing term, which is non-zero only in the last component.

Following the notation introduced in §2c, let us begin with the time series $x(t)$ for $t = 0$, $\Delta t, \dots, (q - 1)\Delta t$. We construct a time-delay embedding $H \in \mathbb{R}^{m \times n}$, where we assume $m \ll n$.

Next, we compute the SVD of $H$ and show that the singular vectors correspond to the Frenet–Serret frame at a fixed point in time. In particular, to compute the SVD of this matrix, we consider the transpose $H^{\mathsf{T}} \in \mathbb{R}^{n \times m}$, which is also a Hankel matrix. Thus, the columns of $H^{\mathsf{T}}$ can be thought of as a trajectory $h(t) \in \mathbb{R}^n$ for $t = 0, \Delta t, \dots, (m - 1)\Delta t$. For simplicity, we shift the origin of time so that $h(t)$ spans $t = -(m - 1)\Delta t/2, \dots, 0, \dots (m - 1)\Delta t/2$, and we denote $h(i\Delta t)$ as $h_i$. In this form,

$$H^{\mathsf{T}} = \begin{bmatrix} | & \cdots & | & \cdots & | \\ h_{(-m+1)/2} & \cdots & h_0 & \cdots & h_{(m-1)/2} \\ | & \cdots & | & \cdots & | \end{bmatrix}.$$

Subtracting the central column $h_0$ from $H^{\mathsf{T}}$ (or equivalently, the central row of $H$) yields the centred matrix

$$\bar{H}^{\mathsf{T}} = H^{\mathsf{T}} - h_0 \mathbf{1}^{\mathsf{T}}. \tag{3.1}$$

We can then express $h_i$ as a Taylor expansion about $h_0$,

$$h_i - h_0 = h_0' i\Delta t + \frac{1}{2} h_0'' (i\Delta t)^2 + \frac{1}{3!} h_0''' (i\Delta t)^3 + \cdots. \tag{3.2}$$

We note that this is a Taylor expansion in each row of $H^{\mathsf{T}}$. The top right image in figure 3 shows sample rows for the Lorenz system. The images with red lines show the sample rows of $H$ (left)

and sample rows of $H^\mathsf{T}$ (right). Many of these curves look nearly linear so even a low-order Taylor expansion would yield good approximations.

With this in mind, applying the results of [33] described in §2e yields the SVD,[1]

$$
\bar{H}^\mathsf{T} = \underbrace{\begin{bmatrix} | & | & | & \\ e_0^1 & e_0^2 & e_0^3 & \cdots \\ | & | & | & \end{bmatrix}}_{V} \underbrace{\begin{bmatrix} \sigma_1 & & & \\ & \sigma_2 & & \\ & & \sigma_3 & \\ & & & \ddots \end{bmatrix}}_{\Sigma} \underbrace{\begin{bmatrix} - & p_1 & - \\ - & p_3 & - \\ - & p_3 & - \\ & \vdots & \end{bmatrix}}_{U^\mathsf{T}}.
\tag{3.3}
$$

The singular vectors in $V$ correspond to the Frenet–Serret frame (the Gram–Schmidt method applied to the vectors, $h_0', h_0'', h_0'''$),

$$
e_0 = \frac{h_0'}{\|h_0'\|}
$$

and

$$
e_0^i = \frac{h_0^{(i)} - \sum_{j=1}^{i-1} \langle h_0^{(i)}, e_0^j \rangle e_0^j}{\left\| h_0^{(i)} - \sum_{j=1}^{i-1} \langle h_0^{(i)}, e_0^j \rangle e_0^j \right\|}.
$$

The matrix $U$ is similarly defined by the discrete orthogonal polynomials

$$
p_1 = \frac{1}{c_1} p
$$

and

$$
p_i = \frac{1}{c_i} \left( p^i - \sum_{j=1}^{i-1} \langle p^i, p_j \rangle p_j \right),
$$

where $p$ is the vector

$$
p = \begin{bmatrix} \dfrac{(-m+1)}{2} & \dfrac{(-m+2)}{2} & \cdots & 0 & \cdots & \dfrac{(m-2)}{2} & \dfrac{(m-1)}{2} \end{bmatrix},
\tag{3.4}
$$

and where $c_i$ is a normalization constant so that $\langle p_i, p_i \rangle = 1$. Note that $p^i$ here means raise $p$ to the power $i$ element-wise. These polynomials are similar to the discrete orthogonal polynomials defined in [68], except $p$ is the normalized ones vector $1/c_1 [1 \cdots 1]$. These polynomials will be discussed further in §3c.

Next, we build a regression model of the dynamics. We first consider the case where the system is closed (i.e. $\bar{H}$ has rank $r$). By (3.3), $V = [e_0^1 \ e_0^2 \ \cdots]$ well-approximates the Frenet–Serret frame at the fixed point in time $t = 0$. Following the Frenet–Serret equations (2.10),

$$
\frac{dV^\mathsf{T}}{dt} = AV^\mathsf{T},
\tag{3.5}
$$

where $A = \|h_0'\| K$. Here, $K$ is a constant tridiagonal and antisymmetric matrix, which corresponds to the curvatures at $t = 0$. From the dual perspective, we can think about the set of vectors $\{e_0^1, e_0^2, \ldots, e_0^r\}$ as an $r$-dimensional time series over $n$ snapshots in time,

$$
V^\mathsf{T} = \begin{bmatrix} - & v_1(t) & - \\ - & v_2(t) & - \\ & \vdots & \\ - & v_r(t) & - \end{bmatrix} = \begin{bmatrix} - & e_0^1 & - \\ - & e_0^2 & - \\ & \vdots & \\ - & e_0^r & - \end{bmatrix} \in \mathbb{R}^{r \times n}.
\tag{3.6}
$$

Here, $v(t) = [v_1(t), v_2(t), \ldots v_r(t)]^\mathsf{T} \in \mathbb{R}^r$ denotes the $r$-dimensional trajectory, which corresponds to the $r$-dimensional coordinates considered in (2.5) for HAVOK. From (3.5), these dynamics must

---

[1] We define the left singular matrix as $V$ and the right singular matrix as $U$. This definition can be thought of as taking the SVD of the transpose of the matrix $H - 1h_0^\mathsf{T}$. This keeps the definitions of the matrices more in line with the notation used in HAVOK.

therefore satisfy

$$\dot{v}(t) = A v(t),$$

where $A$ is a skew-symmetric tridiagonal matrix. If the system is not closed, the dynamics take the form

$$
\begin{bmatrix} \dot{v}_1 \\ \dot{v}_2 \\ \vdots \\ \dot{v}_r \\ \dot{v}_{r+1} \\ \vdots \end{bmatrix}
= \|h_0'\|
\begin{bmatrix} 0 & \kappa_1 & & & & \\ -\kappa_1 & \ddots & \ddots & & & \\ & \ddots & 0 & \ddots & & \\ & & -\kappa_{r-1} & 0 & \kappa_r & \\ & & & -\kappa_r & 0 & \ddots \\ & & & & \ddots & \ddots \end{bmatrix}
\begin{bmatrix} v_1 \\ v_2 \\ \vdots \\ v_r \\ v_{r+1} \\ \vdots \end{bmatrix}.
$$

We note that, due to the tridiagonal structure of $K$, the governing dynamics of the first $r-1$ coordinates $v_1(t), \dots v_{r-1}(t)$ are the same as in the unforced case. The dynamics of the last coordinate includes an additional term $\dot{v}_r = -\kappa_{r-1}v_{r-1} + \kappa_{r+1}v_{r+1}$. The dynamics therefore take the form,

$$\frac{dv}{dt} = A v(t) + B v_{r+1}(t),$$

where $B$ is a vector that is non-zero only its last coordinate. Thus, we recover a model as in (2.8), but with the desired tridiagonal skew-symmetric structure. The matrix of curvatures is simply given by $K = A / \|h_0'\|$.

To compute $A$, similar to (2.6), we define two time-shifted matrices

$$
V_1^{\mathsf{T}} = \begin{bmatrix} v(t_1) & v(t_2) & \cdots & v(t_{m-1}) \end{bmatrix}
\quad \text{and} \quad
V_2^{\mathsf{T}} = \begin{bmatrix} v(t_2) & v(t_3) & \cdots & v(t_m) \end{bmatrix}.
\tag{3.7}
$$

The matrix $A$ may then be approximated as

$$
A = \frac{dV^{\mathsf{T}}}{dt} V^{\mathsf{T}\dagger} \approx \left( \frac{V_2 - V_1}{\Delta t} \right)^{\mathsf{T}} V_1^{\mathsf{T}\dagger}.
\tag{3.8}
$$

In summary, we have shown here that the trajectories of singular vectors $v(t)$ from a time-delay embedding are governed by approximately tridiagonal antisymmetric dynamics, with a forcing term non-zero only in the last component. Comparing these steps with those described in §2c, we see that the estimation of $K$ is nearly identical to the steps in HAVOK. In particular, $\|h_0\|K$ is the linear dynamics matrix $A$ in HAVOK. The only difference is the centring step in (3.1), which is further discussed in §3c.

Note that unlike in the general case for the Frenet–Serret equations, the dynamics matrix here is constant, a surprising result. This is directly due to the time-delay nature of the data and in particular depends on how well $h$ is approximated by its Taylor expansion in (3.2). These assumptions will be explored in more detail in §4.

## (b) HAVOK computes approximate curvatures in a synthetic example

To illustrate the correspondence between non-zero elements of the HAVOK dynamics matrix and curvatures, we start by considering an analytically tractable synthetic example. We start by applying the steps of HAVOK as described in [5] with an additional centring step. The resultant modes and terms on the sub- and super-diagonals of the dynamics matrix are then compared with curvatures computed with an analytic expression, and we show that they are approximately the same, scaled by a factor of $\|h_0'\|$.

We consider data from the one-dimensional system governed by

$$x(t) = \sin(t) + \sin(2t),$$

for $t \in [0, 10]$ and sampled at $\Delta t = 0.001$. Following HAVOK, we form the time-delay matrix $H \in \mathbb{R}^{41 \times 9961}$ then centre the data, subtracting the middle row $h_0$ from all other rows, which forms $\bar{H}$. We next apply the SVD to $\bar{H}^{\mathsf{T}} = V \Sigma U^{\mathsf{T}}$.

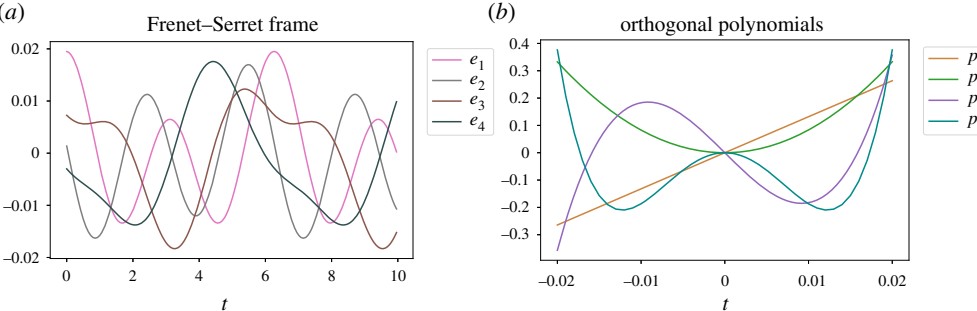

**Figure 4.** Frenet–Serret frame (*a*) and corresponding orthogonal polynomials (*b*) for HAVOK applied to time-series generated by $x(t) = \sin(t) + \sin(2t)$. The orthogonal polynomials and the Frenet–Serret frame are the right singular vectors $\boldsymbol{U}$ and left singular vectors $\boldsymbol{V}$ of $\bar{\boldsymbol{H}}$, respectively.

Figure 4 shows the columns of $\boldsymbol{U} \in \mathbb{R}^{41 \times 4}$ and the columns of $\boldsymbol{V} \in \mathbb{R}^{9961 \times 4}$. The columns of $\boldsymbol{U}$ correspond to the orthogonal polynomials described in §3c and the columns of $\boldsymbol{V}$ are the instantaneous basis vectors $\boldsymbol{e}_i$ for the 9961-dimensional Frenet–Serret frame. To compute the derivative of the state, we now treat $\boldsymbol{V}$ as a four-dimensional trajectory with 9961 snapshots. Applying DMD to $\boldsymbol{V}$ yields the $\boldsymbol{A}$ matrix,

$$A = \begin{bmatrix} -1.245 \times 10^{-3} & 1.205 \times 10^{-2} & 4.033 \times 10^{-6} & 1.444 \times 10^{-7} \\ -1.224 \times 10^{-2} & 3.529 \times 10^{-4} & 4.458 \times 10^{-3} & 2.283 \times 10^{-6} \\ -9.390 \times 10^{-4} & -3.467 \times 10^{-3} & 5.758 \times 10^{-4} & 6.617 \times 10^{-3} \\ 3.970 \times 10^{-4} & -6.568 \times 10^{-4} & -7.451 \times 10^{-3} & 2.835 \times 10^{-4} \end{bmatrix}. \tag{3.9}$$

This matrix is approximately antisymmetric and tridiagonal as we expect.

Next, we compute the Frenet–Serret frame for the time-delay embedding using analytic expressions and show that HAVOK indeed extracts the curvatures of the system multiplied by $\|\boldsymbol{h}_0'\|$. Forming the time-delay matrix, we can easily compute $\boldsymbol{h}_0 = [x_{0.02}, x_{0.02+\Delta t} \dots, x_{9.98}]$.

$$\boldsymbol{h}_0 = \Big[ \sin(t) + \sin(2t) \text{ for } t \in [0.02, 0.021, \dots, 9.98] \Big]$$

and the corresponding derivatives,

$$\dot{\boldsymbol{h}}_0 = \Big[ \cos(t) + 2\cos(2t) \text{ for } t \in [0.02, 0.021, \dots, 9.98] \Big]$$

$$\ddot{\boldsymbol{h}}_0 = \Big[ -\sin(t) - 4\sin(2t) \text{ for } t \in [0.02, 0.021, \dots, 9.98] \Big]$$

$$\dddot{\boldsymbol{h}}_0 = \Big[ -\cos(t) - 8\cos(2t) \text{ for } t \in [0.02, 0.021, \dots, 9.98] \Big]$$

and

$$\boldsymbol{h}_0^{(4)} = \Big[ \sin(t) + 16\sin(2t) \text{ for } t \in [0.02, 0.021, \dots, 9.98] \Big].$$

The fifth derivative $\boldsymbol{h}^{(5)}$ is given by $\cos(t) + 32\cos(2t)$ and can be expressed as a linear combination of the previous derivatives, namely, $\boldsymbol{h}_0^{(5)} = -5\dddot{\boldsymbol{h}}_0 - 4\dot{\boldsymbol{h}}_0$. This can also be shown using the fact that $x(t)$ satisfies the fourth-order ordinary differential equation $x^{(4)} + 5\ddot{x} + 4x = 0$.

Since only the first four derivatives are linearly independent, only the first three curvatures are non-zero. Furthermore, exact values of the first three curvatures can be computed analytically

using the following formulae from [69],

$$\kappa_1 = \frac{\sqrt{\det\left(\begin{bmatrix}\dot{h}_0 & \ddot{h}_0\end{bmatrix}^{\mathsf{T}}\begin{bmatrix}\dot{h}_0 & \ddot{h}_0\end{bmatrix}\right)}}{\|\dot{h}_0\|^{3/2}}, \quad \kappa_2 = \frac{\sqrt{\det\left(\begin{bmatrix}\dot{h}_0 & \ddot{h}_0 & \dddot{h}_0\end{bmatrix}^{\mathsf{T}}\begin{bmatrix}\dot{h}_0 & \ddot{h}_0 & \dddot{h}_0\end{bmatrix}\right)}}{\det\left(\begin{bmatrix}\dot{h}_0 & \ddot{h}_0\end{bmatrix}^{\mathsf{T}}\begin{bmatrix}\dot{h}_0 & \ddot{h}_0\end{bmatrix}\right)},$$

$$\kappa_3 = \frac{\sqrt{\det\left(\begin{bmatrix}\dot{h}_0 & \ddot{h}_0 & \dddot{h}_0 & h_0^{(4)}\end{bmatrix}^{\mathsf{T}}\begin{bmatrix}\dot{h}_0 & \ddot{h}_0 & \dddot{h}_0 & h_0^{(4)}\end{bmatrix}\right)}\det\left(\begin{bmatrix}\dot{h}_0 & \ddot{h}_0\end{bmatrix}^{\mathsf{T}}\begin{bmatrix}\dot{h}_0 & \ddot{h}_0\end{bmatrix}\right)}{\det\left(\begin{bmatrix}\dot{h}_0 & \ddot{h}_0 & \dddot{h}_0\end{bmatrix}^{\mathsf{T}}\begin{bmatrix}\dot{h}_0 & \ddot{h}_0 & \dddot{h}_0\end{bmatrix}\right)\|\dot{h}_0\|}.$$

These formulae yield the values $\kappa_1 = 1.205 \times 10^{-2}$, $\kappa_2 = 4.46 \times 10^{-3}$ and $\kappa_3 = 6.62 \times 10^{-3}$, respectively.

As expected, these curvature values are very close to those computed with HAVOK, highlighted in (3.9). In particular, the super-diagonal entries of the matrix appear to be very good approximations to the curvatures. The reasons why the super-diagonal, but not the sub-diagonal, is so close in value to the true curvatures is not yet well understood. Furthermore, in §5, we use the theoretical insights from §3a to propose a modification to the HAVOK algorithm that yields an even better approximation to curvatures in the Frenet–Serret frame.

## (c) Orthogonal polynomials and centring

In the decomposition in (3.3), we define a set of orthonormal polynomials. Here, we discuss the properties of these polynomials, comparing them with the Legendre polynomials and providing explicit expressions for the first several terms in this series.

In §3a, we apply the SVD to the centred matrix $\bar{H}$, as in (3.3). The columns of $U$ in this decomposition yield a set of orthonormal polynomials, which are defined by (2.14). In the continuous case, the inner product in (2.14) is $\langle a(t), b(t)\rangle = \int_{-p}^{p} a(t)b(t)dt$, while in the discrete case $\langle a, b\rangle = \sum_{j=-p}^{p} a_j b_j$. The first five polynomials in the discrete case may be found in the electronic supplementary material, Note 1. The first five of these polynomials $p_i(x)$ in the continuous case are

$$p_1(x) = \frac{x}{c_1(p)}, \quad \text{where } c_1(p) = \frac{\sqrt{6}\sqrt{p^3}}{3}$$

$$p_2(x) = \frac{x^2}{c_2(p)}, \quad \text{where } c_2(p) = \frac{\sqrt{10}\sqrt{p^5}}{5}$$

$$p_3(x) = \frac{1}{c_3(p)}\left(x^3 - \frac{3}{5}p^2 x\right), \quad \text{where } c_3(p) = \frac{2\sqrt{14}\sqrt{p^7}}{35}$$

$$p_4(x) = \frac{1}{c_4(p)}\left(x^4 - \frac{5}{7}p^2 x^2\right), \quad \text{where } c_4(p) = \frac{2\sqrt{2}\sqrt{p^9}}{21}$$

and
$$p_5(x) = \frac{1}{c_5(p)}\left(x^5 + \frac{5}{21}p^4 x - \frac{10}{9}p^2 x^3\right), \quad \text{where } c_5(p) = \frac{8\sqrt{22}\sqrt{p^{11}}}{693}.$$

By construction, $p_i(t)$ form a set of orthonormal polynomials, where $p_i(t)$ has degree $i$.

Interestingly, these orthogonal polynomials are similar to the Legendre polynomials $l_i$ [70,71], which are defined by the recursive relation

$$l_1 = \frac{1}{c_1}\begin{bmatrix} 1 & 1 & \cdots & 1 \end{bmatrix}$$

and

$$l_i = \frac{1}{p_i}\left(\boldsymbol{p}^i - \sum_{k=1}^{i-1}\langle \boldsymbol{p}^i, l_k\rangle\right),$$

where $\boldsymbol{p}$ is as defined in (3.4). For the corresponding Legendre polynomials normalized over $[-p, p]$, we refer the reader to [68].

The key difference between these two sets of polynomials is that the first polynomial $\boldsymbol{p}_1$ is linear, while the first Legendre polynomial is constant (i.e. corresponding in the discrete case to the normalized ones vector). In particular, if $\boldsymbol{H}$ is not centred before decomposition by SVD, the resulting columns of $\boldsymbol{U}$ will be the Legendre polynomials. However, without centring, the resulting $\boldsymbol{V}$ will no longer be the Frenet–Serret frame. Instead, the resulting frame corresponds to applying the Gram–Schmidt method to the set $\{\boldsymbol{\gamma}(t), \boldsymbol{\gamma}'(t), \boldsymbol{\gamma}''(t), \ldots\}$ instead of $\{\boldsymbol{\gamma}'(t), \boldsymbol{\gamma}''(t), \boldsymbol{\gamma}'''(t), \ldots\}$. Recently, it has been shown that using centring as a preprocessing step is beneficial for the DMD [72]. That being said, since the derivation of the tridiagonal and antisymmetric structure seen in the Frenet–Serret frame is based on the properties of the derivatives and orthogonality, this same structure can be computed without the centring step.

## 4. Limits and requirements

Section 3a has shown how HAVOK yields a good approximation to the Frenet–Serret frame in the limit that the time interval spanned by each row of $\boldsymbol{H}$ goes to zero. To be more precise, HAVOK yields the Frenet–Serret frame if (2.13) is satisfied. However, this property can be difficult to check in practice. Here, we establish several rules for choosing and structuring the data so that the HAVOK dynamics matrix adopts the structure we expect from theory.

**Choose $\Delta t$ to be small.** The specific constraint we have from (2.13) is

$$\|\boldsymbol{h}_0' t_i\| \gg \left\|\frac{\boldsymbol{h}_0''}{2}t_i^2\right\| \gg \left\|\frac{\boldsymbol{h}_0'''}{6}t_i^3\right\| \gg \cdots \gg \left\|\frac{\boldsymbol{h}_0^{(k)}}{k!}t_i^k\right\|,$$

for $-m\Delta t/2 \le t_i \le m\Delta t/2$ or more simply $|t_i| \le m\Delta t$, where $\Delta t$ is the sampling period (inverse of the sampling frequency) of the data and $m$ is the number of delays in the Hankel matrix $\boldsymbol{H}$. If we assume that $m\Delta t < 1$, then rearranging,

$$m\Delta t \ll \frac{2\|\boldsymbol{h}_0'\|}{\|\boldsymbol{h}_0''\|}, \frac{3\|\boldsymbol{h}_0''\|}{\|\boldsymbol{h}_0'''\|}, \ldots, \frac{k\|\boldsymbol{h}_0^{(k-1)}\|}{\|\boldsymbol{h}_0^{(k)}\|}. \tag{4.1}$$

In practice, since the series of ratios of derivatives defined in (4.1) grows, it is only necessary to check the first inequality. By choosing the sampling period of the data to be small, we can constrain the data to satisfy this inequality. To illustrate the effect of decreasing $\Delta t$, figure 5a–d shows the dynamics matrices $\boldsymbol{A}$ computed by the HAVOK algorithm for the Lorenz system for a fixed number of rows of data and fixed time span of the simulation. As $\Delta t$ becomes smaller, $\boldsymbol{A}$ becomes more structured in that it is antisymmetric and tridiagonal.

**Choose the number of columns $n$ to be large.** The number of columns comes into the Taylor expansion through the derivatives $\|\boldsymbol{h}_0^{(k)}\|$, since $\boldsymbol{h}_0^{(k)} \in \mathbb{R}^n$.

For the synthetic example $x(t) = \sin(t) + 2\sin(t)$, we can show that the ratio $2\|\boldsymbol{h}_0'\|/\|\boldsymbol{h}_0''\|$ saturates to a fixed value in the limit as $n$ goes to infinity (see the electronic supplementary material, Note 2). However, for short time series (small values of $n$), this ratio can be arbitrarily small, and hence (4.1) will be difficult to satisfy.

We illustrate this in figure 5 using data from the Lorenz system. We compute and plot the HAVOK linear dynamics matrix for a varying number of columns $n$, while fixing the sampling frequency and number of rows $m$. We see that as we increase the number of columns, the dynamics becomes more skew-symmetric and tridiagonal. In general, due to practical constraints and restrictions, it may be difficult to guarantee that given data satisfies these two requirements. In §§4a and 5, we propose methods to tackle this challenge.

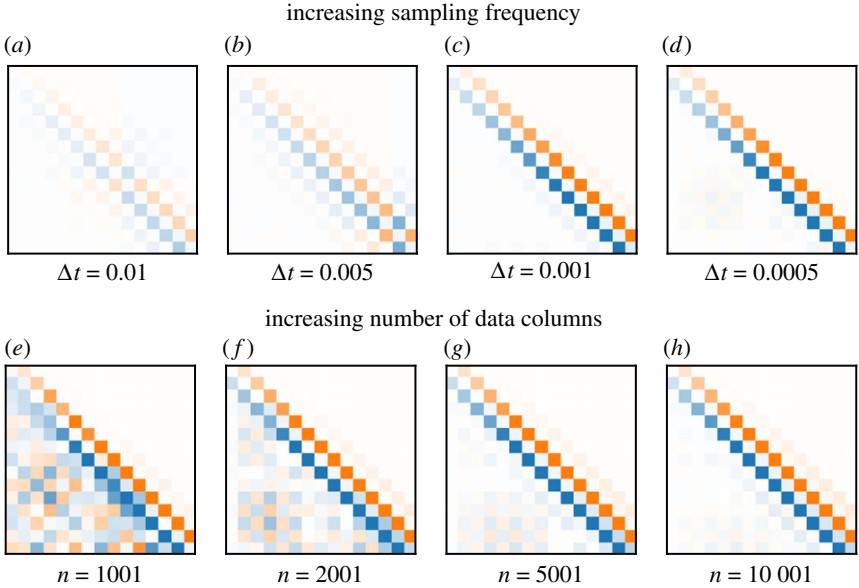

**Figure 5.** Increasing sampling frequency and number of columns yields more structured HAVOK models for the Lorenz system. Given the Hankel matrix **H**, the linear dynamical model is plotted for values of sampling period $\Delta t$ equal to 0.01, 0.005, 0.001, 0.0005 for a fixed number of rows and fixed time span of measurement (*a*–*d*). Similarly, the model is plotted for values of the number of columns *n* equal to 1001, 2001, 5001 and 10 001 for fixed sampling frequency and number of delays *m* (*e*–*h*). As we increase the sampling frequency and the number of columns of the data, **A** becomes more antisymmetric with non-zero elements only on the super- and sub-diagonals. These trends illustrate the results in §4. (Online version in colour.)

## (a) Interpolation

From the first requirement, we see that the sampling frequency $\Delta t$ needs to be sufficiently small to recover the antisymmetric structure in *A*. However, in practice, it is not always possible to satisfy this sampling criterion.

One solution to remedy this is to use data interpolation. To be precise, we can increase the sampling rate by spline interpolation, then construct *H* from the interpolated data that satisfies (4.1). The ratio of the derivatives $\|h_0'\|/\|h_0''\|$, $\|h_0''\|/\|h_0'''\|$, ... may also contain some dependence on $\Delta t$, but we observe that this dependence is not significantly affected in practice.

As an example, we consider a set of time-series measurements generated from the Lorenz system (see §5 for more details about this system). We start with a sampling period of $\Delta t = 0.1$ (figure 6*a*–*c*). Note that here we have simulated the Lorenz system at high temporal resolution then subsampled to produce these time-series data. Applying HAVOK with centring and $m = 201$, we see that *A* is not antisymmetric and the columns of *U* are not the orthogonal polynomials like in the synthetic example shown in figure 4.

Next, we apply cubic spline interpolation to these data, evaluating at a sampling rate of $\Delta t = 0.001$ (figure 6*d*–*f*). We note that, especially for real-world data with measurement noise, this interpolation procedure also serves to smooth the data, making the computation of its derivatives more tractable [73]. Applying HAVOK to this interpolated data yields a new antisymmetric *A* matrix and the *U* corresponds to the orthogonal polynomials described in §3c.

## 5. Promoting structure in the HAVOK decomposition

HAVOK yields a linear model of a dynamical system explained by the Frenet–Serret frame, and by leveraging these theoretical connections, here we propose a modification of the HAVOK

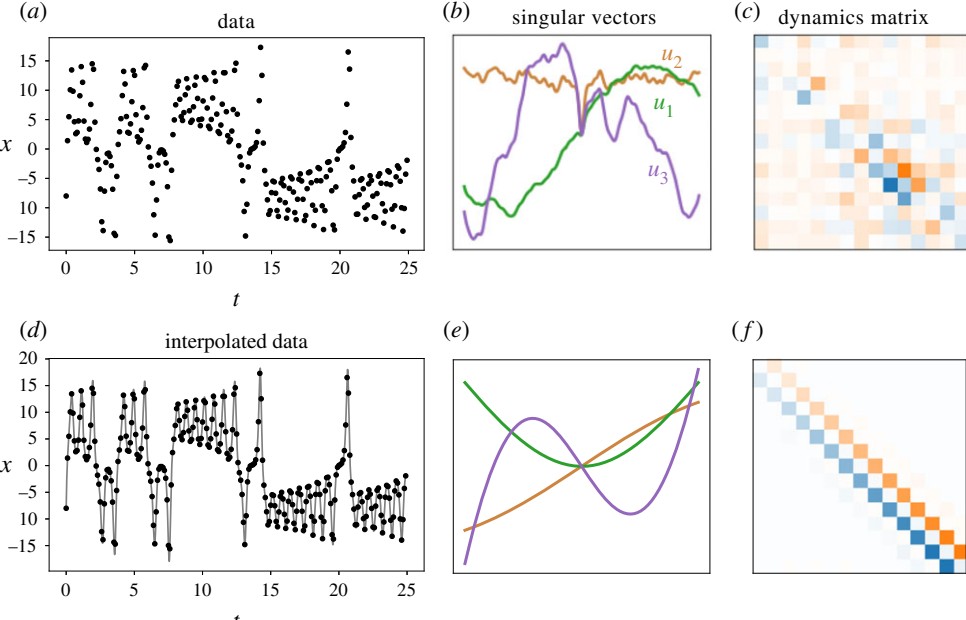

**Figure 6.** In the case where a dynamical system is sparsely sampled, interpolation can be used to recover a more tridiagonal and antisymmetric matrix for the linear model in HAVOK. First, we simulate the Lorenz system, measuring $x(t)$ with a sampling period of $\Delta t = 0.1$. The resulting dynamics model $\boldsymbol{A}$ and corresponding singular vectors of $\boldsymbol{U}$ are plotted. Due to the low sampling frequency, these values do not satisfy the requirements in (4.1). Consequently, the dynamics matrix is not antisymmetric and the singular vectors do not correspond to the orthogonal polynomials in §3c. Next, the data are interpolated using cubic splines and subsequently sampled using a sampling period of $\Delta t = 0.001$. In this case, the data satisfy the assumptions in (4.1), which yields the tridiagonal antisymmetric structure for $\boldsymbol{A}$ and orthogonal polynomials for $\boldsymbol{U}$ as predicted. (Online version in colour.)

algorithm to promote this antisymmetric structure. We refer to this algorithm as sHAVOK and describe it in §5a. Compared with HAVOK, sHAVOK yields structured dynamics matrices that better approximate the Frenet–Serret frame and more closely estimate the curvatures. Importantly, sHAVOK also produces better models of the system using significantly less data. We demonstrate its application to three nonlinear synthetic example systems in §5b and two real-world datasets in §5c.

## (a) The sHAVOK algorithm

We propose a modification to the HAVOK algorithm that more closely induces the antisymmetric structure in the dynamics matrix, especially for shorter time series. The key innovation in sHAVOK is the application of two SVDs applied separately to time-shifted Hankel matrices (compare figures 1 and 7). This simple modification enforces that the singular vector bases on which the dynamics matrix is computed are orthogonal, and thus more closely approximate the Frenet–Serret frame.

Building on the HAVOK algorithm as summarized in §2c, we focus on the step where the singular vectors $V$ are split into $V_1$ and $V_2$. In the Frenet–Serret framework, we are interested in the evolution of the orthonormal frame $e_1(t), e_2(t), \ldots, e_r(t)$. In HAVOK, $V_1$ and $V_2$ correspond to instances of this orthonormal frame.

Although $V$ is a unitary matrix, $V_1$ and $V_2$—which each consist of removing a column from $V$—are not. To enforce this orthogonality, we propose to split $\bar{H}$ into two time-shifted matrices $\bar{H}_1$

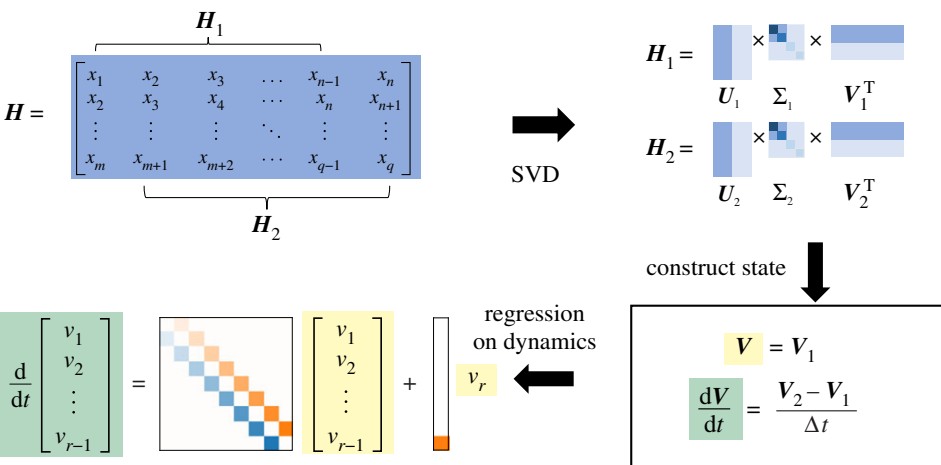

**Figure 7.** Outline of steps in structured HAVOK (sHAVOK). First, given a dynamical system a single variable $x(t)$ is measured. Time-shifted copies of $x(t)$ are stacked to form a Hankel matrix $H$. $H$ is split into two time-shifted matrices, $H_1$ and $H_2$. The singular value decomposition (SVD) is applied to these two matrices individually. This results in reduced order representations, $V_1$ and $V_2$, of $H_1$ and $H_2$, respectively. The matrices, $V_1$ and $V_2$ are then used to construct an approximation to this low-dimensional state and its derivative. Finally, linear regression is performed on these two matrices to form a linear dynamical model with an additional forcing term in the last component. (Online version in colour.)

and $\bar{H}_2$ (figure 7) and then compute two SVDs with rank truncation $r$,

$$\bar{H}_1 = U_1 \Sigma_1 V_1^\mathsf{T} \quad \text{and} \quad \bar{H}_2 = U_2 \Sigma_2 V_2^\mathsf{T}.$$

By construction, $V_1$ and $V_2$ are now orthogonal matrices.

Like in HAVOK, our goal is to estimate the dynamics matrix $A$ such that

$$\dot{v}(t) = A v(t).$$

To do so, we use the matrices $V_1$ and $V_2$ to construct the state and its derivative,

$$V = V_1$$

and

$$\frac{\mathrm{d}V}{\mathrm{d}t} = \frac{V_2 - V_1}{\Delta t}.$$

$A$ then satisfies

$$A = \frac{\mathrm{d}V}{\mathrm{d}t}^\mathsf{T} V^{\mathsf{T}\dagger} = \left(\frac{V_2 - V_1}{\Delta t}\right)^\mathsf{T} V_1^{\dagger} = \left(\frac{V_2^\mathsf{T} V_1 - I}{\Delta t}\right). \tag{5.1}$$

If this system is not closed (non-zero forcing term), then $V_2$ is defined as columns 2 to $n-1$ of the SVD singular vectors with an $r-1$ rank truncation $V_{r-1}^\mathsf{T}$, and $A \in \mathbb{R}^{r-1 \times r-1}$ and $B \in \mathbb{R}^{r-1 \times 1}$ are computed as $[A, B] = (V_2^\mathsf{T} V_1 - I)/\Delta t$. The corresponding pseudocode is elaborated in the electronic supplementary material, Note 3. We note that sHAVOK requires one additional SVD evaluation compared with HAVOK. For situations in which runtime is a limiting factor, $\bar{H}_2$ may be expressed using rank one updates of $\bar{H}_1$. Using this factor, efficient methods may be leveraged to compute the SVD of $\bar{H}_2$ from $\bar{H}_1$, with a negligible increase to runtime [74,75].

As a simple analytic example, we apply sHAVOK to the same system described in §b generated by $x(t) = \sin(t) + \sin(2t)$. The resulting dynamics matrix is

$$A = \begin{bmatrix} -1.116 \times 10^{-5} & 1.204 \times 10^{-2} & -1.227 \times 10^{-5} & 8.728 \times 10^{-8} \\ -1.204 \times 10^{-2} & -1.269 \times 10^{-5} & 4.458 \times 10^{-3} & 4.650 \times 10^{-6} \\ 2.053 \times 10^{-5} & -4.458 \times 10^{-3} & -4.897 \times 10^{-6} & 6.617 \times 10^{-3} \\ -9.956 \times 10^{-8} & -1.118 \times 10^{-7} & -6.617 \times 10^{-3} & -3.368 \times 10^{-6} \end{bmatrix}.$$

We see immediately that, with this small modification, $A$ has become much more structured compared with (3.9). Specifically, the estimates of the curvatures both below and above the diagonal are now equal, and the rest of the elements in the matrix, which should be zero, are almost all smaller by an order of magnitude. In addition, the curvatures are equal to the true analytic values up to three decimal places.

We emphasize that from a theoretical standpoint, sHAVOK aligns much more closely with the findings of §3. In particular, sHAVOK enforces that the singular vector bases on which the dynamics matrix is computed are orthogonal, and thus more closely approximate the Frenet–Serret frame compared with HAVOK. Methods with stronger theoretical foundations are beneficial as they allow us to (1) better predict/understand their behaviour on new datasets and (2) more easily understand their underlying assumptions and areas for future modifications. For further analysis of the sHAVOK method for varying lengths of data, initial conditions, rank truncations and noise levels, see the electronic supplementary material, Note §5–8.

## (b) Comparison of HAVOK and sHAVOK for three synthetic examples

The results of HAVOK and sHAVOK converge in the limit of infinite data,[2] and the models they produce are most different in cases of shorter time-series data, where we may not have measurements over long periods of time. Using synthetic data from three nonlinear example systems, we compute models using both methods and compare the corresponding dynamics matrices $A$ (figure 8). In every case, the $A$ matrix computed using the sHAVOK algorithm is more antisymmetric and has a stronger tridiagonal structure than the corresponding matrix computed using HAVOK.

In addition to the dynamics matrices, we also show in figure 8 the eigenvalues of $A$, $\omega_k \in \mathbb{C}$ for $k = 1, \dots r$ for HAVOK (teal) and sHAVOK (maroon). We additionally plot the eigenvalues (black crosses) corresponding to those computed from the data measured in the large data limit, but at the same sampling frequency. In this large data limit, both sHAVOK and HAVOK yield the same antisymmetric tridiagonal dynamics matrix and corresponding eigenvalues. Comparing the eigenvalues, we immediately see that eigenvalues from sHAVOK more closely match those computed in the large data limit. Thus, even with a short trajectory, we can still recover models and key features of the underlying dynamics.

We emphasize here that sHAVOK is robust to initial conditions. In particular, for the first example, corresponding to the Lorenz system, we plot the HAVOK and sHAVOK results for three different subsets of the data. In all of these cases, although the HAVOK dynamics matrix varies significantly in structure, the sHAVOK matrix remains antisymmetric and tridiagonal. Furthermore, the sHAVOK eigenvalues are much closer to those from the long trajectory compared with HAVOK. We describe each of the systems and their configurations below.

**Lorenz attractor:** We first illustrate these two methods on the Lorenz system. Originally developed in the fluids community, the Lorenz [76] system is governed by three first-order differential equations [76]:

$$\dot{x} = \sigma(y - x),$$

$$\dot{y} = x(\rho - z) - y$$

and

$$\dot{z} = xy - \beta z.$$

---

[2]See the electronic supplementary material, Note §4, for more details.

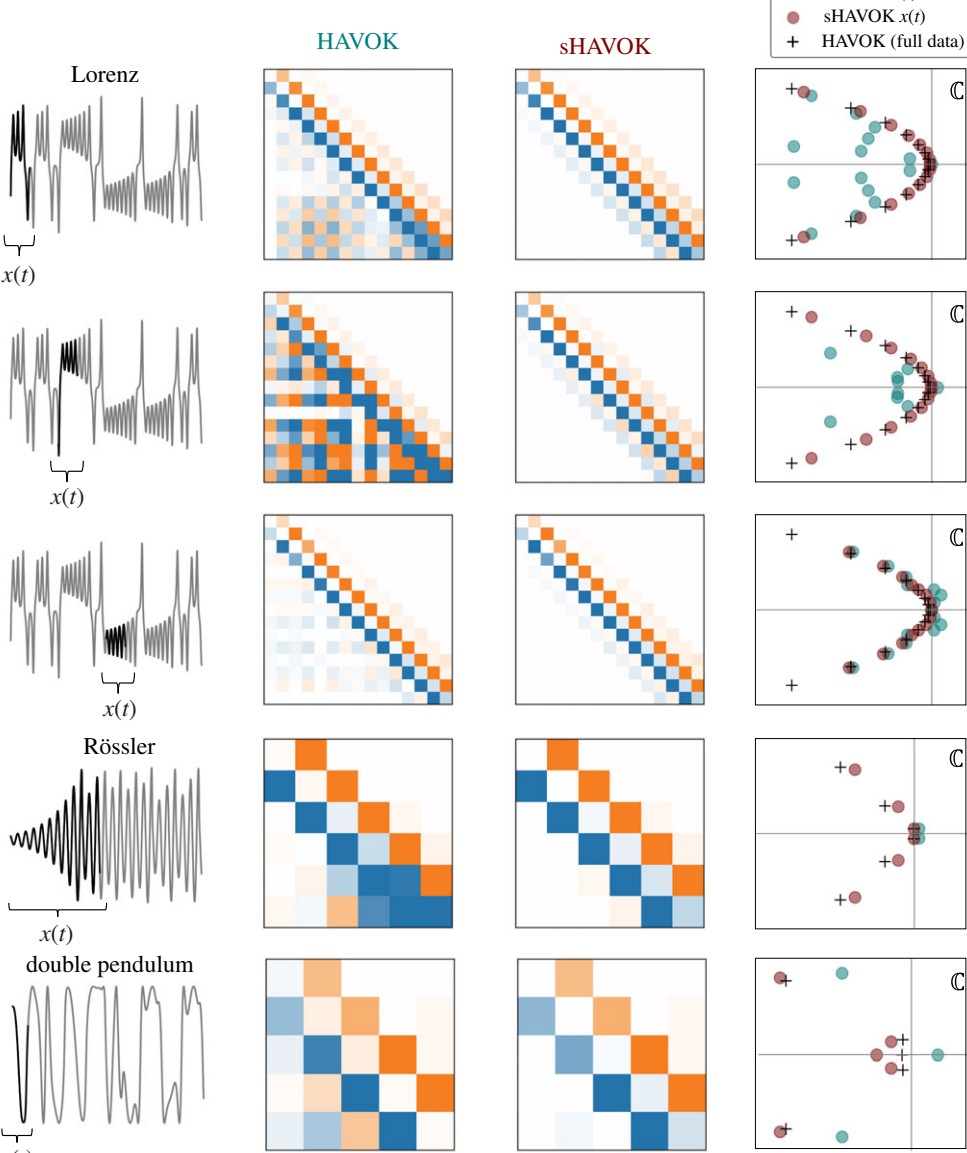

**Figure 8.** Structured HAVOK (sHAVOK) yields more structured models from short trajectories than HAVOK. For each system, we simulated a trajectory extracting a single coordinate in time (grey). We then apply HAVOK and sHAVOK to data $x(t)$ from a short subset of this trajectory, shown in black. The middle columns show the resulting dynamics matrices **A** from the models. The top three rows correspond to different subsets of the Lorenz system, while the fourth and fifth rows correspond to trajectories from the Rössler system and a double pendulum, respectively. Compared with HAVOK, the resulting models for sHAVOK consistently show stronger structure in that they are antisymmetric with non-zero elements only along the sub- and super-diagonals. The corresponding eigenvalue spectra of **A** for HAVOK and sHAVOK are plotted in teal and maroon, respectively, in addition to eigenvalues from HAVOK for the full (grey) trajectory. In all cases, the sHAVOK eigenvalues are much closer in value to those from the long trajectory limit than HAVOK. (Online version in colour.)

The Lorenz system has since been used to model systems in a wide variety of fields, including chemistry [77], optics [78] and circuits [79].

We simulate 3000 samples with initial condition $[-8, 8, 27]$ and a stepsize of $\Delta t = 0.001$, measuring the variable $x(t)$. We use the common parameters $\sigma = 10$, $\rho = 28$ and $\beta = 8/3$. This

trajectory is shown in figure 8 and corresponds to a few oscillations about a fixed point. We choose the lengths of these datasets to be short enough that the HAVOK dynamics matrix is visually neither antisymmetric nor tridiagonal. We compare the spectra with that of a longer trajectory containing 300 000 samples, which we take to be an approximation of the true spectrum of the system.

**Rössler attractor:** The Rössler attractor is given by the following nonlinear differential equations [80,81]:

$$\dot{x} = -y - z,$$
$$\dot{y} = x + ay$$
and
$$\dot{z} = b + z(x - c).$$

We choose to measure the variable $x(t)$. This attractor is a canonical example of chaos, like the Lorenz attractor. Here, we perform a simulation with 70 000 samples and a stepsize of $\Delta t = 0.001$. We choose the following common values of $a = 0.1$, $b = 0.1$ and $c = 14$ and the initial condition $x_0 = y_0 = z_0 = 1$. We similarly plot the trajectory and dynamics matrices. We compare the spectra in this case with a longer trajectory using a simulation for 300 000 samples.

**Double pendulum:** The double pendulum is a similar nonlinear differential equation, which models the motion of a pendulum that is connected at the end to another pendulum [82]. This system is typically represented by its Lagrangian,

$$\mathcal{L} = \frac{1}{6}ml^2(\dot{\theta}_2^2 + 4\dot{\theta}_1^2 + 3\dot{\theta}_1\dot{\theta}_2\cos(\theta_1 - \theta_2)) + \frac{1}{2}mgl(3\cos\theta_1 + \cos\theta_2), \tag{5.2}$$

where $\theta_1$ and $\theta_2$ are the angles between the top and bottom pendula and the vertical axis, respectively. $m$ is the mass at the end of each pendulum, $l$ is the length of each pendulum and $g$ is the acceleration constant due to gravity. Using the Euler–Lagrange equations,

$$\frac{d}{dt}\frac{\partial\mathcal{L}}{\partial\dot{\theta}_i} - \frac{\partial\mathcal{L}}{\partial\theta_i} = 0 \quad \text{for } i = 1, 2,$$

we can construct two second-order differential equations of motion.

The trajectory is computed using a variational integrator to approximate

$$\delta\int_a^b \mathcal{L}(\theta_1, \theta_2, \dot{\theta}_1, \dot{\theta}_2)\,dt = 0.$$

We simulate this system with a stepsize of $\Delta t = 0.001$ and for 1200 samples. We choose $m_1 = m_2 = l_1 = l_2 = 1$ and $g = 10$, and use initial conditions $\theta_1 = \theta_2 = \pi/2$, $\dot{\theta}_1 = -0.01$ and $\dot{\theta}_2 = -0.005$. As our measurement for HAVOK and sHAVOK, we use $x(t) = \sin(\theta_1(t))$ and compare our data with a long trajectory containing 100 000 samples.

## (c) sHAVOK applied to real-world datasets

Here, we apply sHAVOK to two real-world time-series datasets, the trajectory of a double pendulum and measles outbreak data. Similar to the synthetic examples, we find that the dynamics matrix from sHAVOK is much more antisymmetric and tridiagonal compared with the dynamics matrix for HAVOK. In both cases, some of the HAVOK eigenvalues contain positive real components; in other words, these models have unstable dynamics. However, the sHAVOK spectra do not contain positive real components, resulting in much more accurate and stable models (figure 9).

**Double pendulum:** We first look at measurements of a double pendulum [83]. A picture of the set-up can be found in figure 9. The Lagrangian in this case is very similar to that in (5.2). One key difference in the synthetic case is that all of the mass is contained at the joints, while in this experiment, the mass is spread over each arm. To accommodate this, the Lagrangian can be

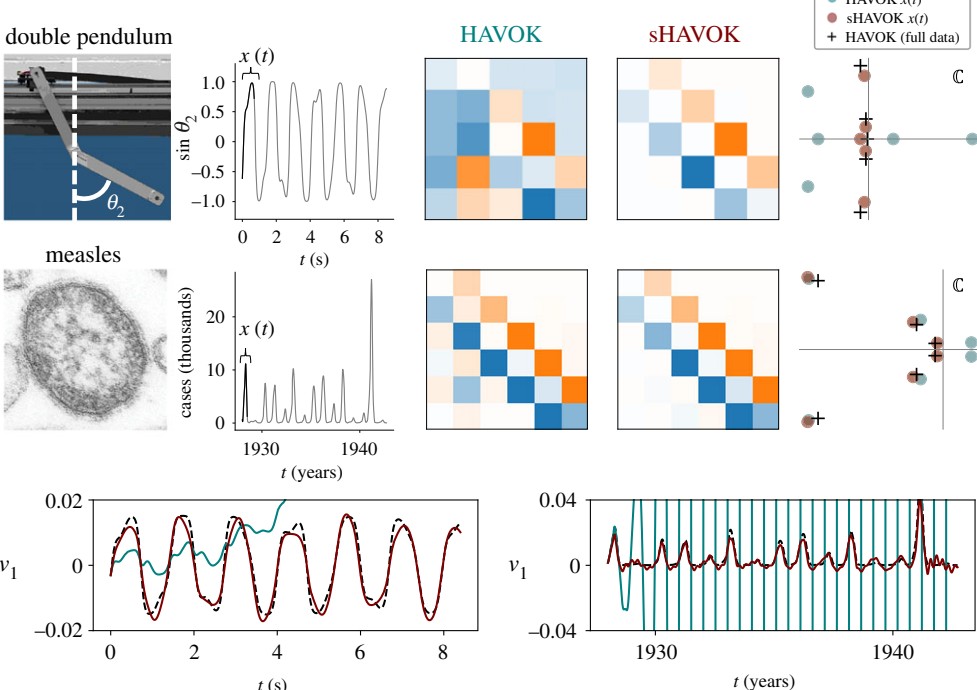

**Figure 9.** Comparison of HAVOK and structured HAVOK (sHAVOK) for two real-world systems: a double pendulum and measles outbreak data. For each system, we measure a trajectory extracting a single coordinate (grey). We then apply HAVOK and sHAVOK to a subset of this trajectory, shown in black. The **A** matrices for the resulting linear dynamical models are shown. sHAVOK yields models with an antisymmetric structure, with non-zero elements only along the sub-diagonal and super-diagonal. The corresponding eigenvalue spectra for HAVOK and sHAVOK are additionally plotted in teal and maroon, respectively, along with eigenvalues from HAVOK for a long trajectory. In both cases, the eigenvalues of sHAVOK are much closer in value to those in the long trajectory limit than HAVOK. Some of the eigenvalues of HAVOK are unstable and have positive real components. The corresponding reconstructions of the first singular vector of the corresponding Hankel matrices are shown along with the real data. Note that the HAVOK models are unstable, growing exponentially due to the unstable eigenvalues, while the sHAVOK models do not. Credit for images on left: (double pendulum) [83] and (measles) CDC/Cynthia S. Goldsmith; William Bellini, PhD. (Online version in colour.)

slightly modified,

$$\mathcal{L} = \frac{1}{2}(m_1(\dot{x}_1^2 + \dot{y}_1^2) + m_2(\dot{x}_2^2 + \dot{y}_2^2)) + \frac{1}{2}(I_1\dot{\theta}_1^2 + I_2\dot{\theta}_2^2) - (m_1y_1 + m_2y_2)g,$$

where $x_1 = a_1 \sin(\theta_1)$, $x_2 = l_1 \sin(\theta_1) + a_2 \sin(\theta_2)$, $y_1 = a_1 \cos(\theta_1)$ and $y_2 = l_1 \cos(\theta_1) + a_2 \cos(\theta_2)$. $m_1$ and $m_2$ are the masses of the pendula, $l_1$ and $l_2$ are the lengths of the pendula, $a_1$ and $a_2$ are the distances from the joints to the centre of masses of each arm, and $I_1$ and $I_2$ are the moments of inertia for each arm. When $m_1 = m_2 = m$, $a_1 = a_2 = l_1 = l_2$ and $I_1 = I_2 = ml^2$ we recover (5.2). We sample the data at $\Delta t = 0.001$ s and plot $\sin(\theta_2(t))$ over a 15 s time interval. The data over this interval appear approximately periodic.

**Measles outbreaks:** As a second example, we apply measles outbreak data from New York City between 1928 and 1964 [84]. The case history of measles over time has been shown to exhibit chaotic behaviour [85,86], and [5] applied HAVOK to measles data and successfully showed that the method could extract transient behaviour.

For both systems, we apply sHAVOK to a subset of the data corresponding to the black trajectories $x(t)$ shown in figure 9. We then compare that with HAVOK applied over the same interval. We use $m = 101$ delays with a $r = 5$ rank truncation for the double pendulum, and

$m = 51$ delays and a $r = 6$ rank truncation for the measles data. For the measles data, prior to applying sHAVOK and HAVOK, the data is first interpolated and sampled at a rate of $\Delta t = 0.0018$ years. Like in previous examples, the resulting sHAVOK dynamics is tridiagonal and antisymmetric while the HAVOK dynamics matrix is not. Next, we plot the corresponding spectra for these two methods, in addition to the eigenvalues applied to HAVOK over the entire time series. Most noticeably, the eigenvalues from sHAVOK are closer to the long data limit values. In addition, two of the HAVOK eigenvalues lie to the right of the real axis, and thus have positive real components. All of the sHAVOK eigenvalues, on the other hand, have negative real components. This difference is most prominent in the reconstructions of the first singular vector. In particular, since two of the eigenvalues from HAVOK are positive, the reconstructed time series grows exponentially. By contrast, for sHAVOK the corresponding time-series remains bounded providing a much better model of the true data.

# 6. Discussion

In this paper, we describe a new theoretical connection between models constructed from time-delay embeddings, specifically using the HAVOK approach, and the Frenet–Serret frame from differential geometry. This unifying perspective explains the peculiar antisymmetric, tridiagonal structure of HAVOK models: namely, the sub- and super-diagonal entries of the linear model correspond to the intrinsic curvatures in the Frenet–Serret frame. Inspired by this theoretical insight, we develop an extension we call *structured* HAVOK that effectively yields models with this structure. Importantly, we demonstrate that this modified algorithm improves the stability and accuracy of time-delay embedding models, especially when data are noisy and limited in length. All code is available at https://github.com/sethhirsh/sHAVOK.

Establishing theoretical connections between time-delay embedding, dimensionality reduction and differential geometry opens the door for a wide variety of applications and future work. By understanding this new perspective, we now better understand the requirements and limitations of HAVOK and have proposed simple modifications to the method which improve its performance on data. However, the full implications of this theory remain unknown. Differential geometry, dimensionality reduction and time-delay embeddings are all well-established fields, and by understanding these connections we can develop more robust and interpretable methods for modelling time series.

For instance, by connecting HAVOK to the Frenet–Serret frame, we recognize the importance of enforcing orthogonality for $V_1$ and $V_2$ and inspired development of sHAVOK. With this theory, we can incorporate further improvements on the method. For example, sHAVOK can be thought of as a first-order forward difference method, approximating the derivative and state by $(V_2 - V_1)/\Delta t$ and $V_1$, respectively. By employing a central difference scheme, such as approximating the state by $V$, we have observed this to further enforce the antisymmetry in the dynamics matrix and move the corresponding eigenvalues towards the imaginary axis.

Throughout this analysis, we have focused purely on linear methods. In recent years, nonlinear methods for dimensionality reduction, such as autoencoders and diffusion maps, have gained popularity [7,87,88]. Nonlinear models similarly benefit from promoting sparsity and interpretability. By understanding the structures of linear models, we hope to generalize these methods to create more accurate and robust methods that can accurately model a greater class of functions.

Data accessibility. All code used to reproduce results in the figures is openly available at https://github.com/sethhirsh/sHAVOK.

Authors' contributions. S.M.H. conceived of the study, designed the analyses, carried out the analyses and wrote the manuscript. S.M.I. helped carry out the computational analyses. S.L.B. helped design the analyses and write the manuscript. J.N.K. helped design the analyses and revised the manuscript. B.W.B. helped design the analyses, coordinate the study and write the manuscript.

Competing interests. The authors have no competing interests to declare.

**Funding.** This work was funded by the Army Research Office (W911NF-17-1-0306 to S.L.B.); Air Force Office of Scientific Research (FA9550-17-1-0329 to J.N.K.); the Air Force Research Laboratory (FA8651-16-1-0003 to B.W.B.); the National Science Foundation (award no. 1514556 to B.W.B.); the Alfred P. Sloan Foundation and the Washington Research Foundation to B.W.B.

**Acknowledgements.** We are grateful for discussions with S. H. Singh and K. D. Harris; and to K. Kaheman for providing the double pendulum dataset. We especially thank A. G. Nair for providing valuable insights and feedback in designing the analysis.

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
