## [Peer Review File · Proceedings. Mathematical, Physical, and Engineering Sciences]

Review History

RSPA-2021-0097.R0 (Original submission)

Review form: Referee 1

Is the manuscript an original and important contribution to its field?

Good

Is the paper of sufficient general interest?

Good

Is the overall quality of the paper suitable?

Good

Can the paper be shortened without overall detriment to the main message?

Yes

Do you think some of the material would be more appropriate as an electronic appendix?

No

Do you have any ethical concerns with this paper?

No

Recommendation?

Major revision is needed (please make suggestions in comments)

Comments to the Author(s)

Please see attachment.

Review form: Referee 2**Is the manuscript an original and important contribution to its field?**

Acceptable

Is the paper of sufficient general interest?

Acceptable

Is the overall quality of the paper suitable?

Good

Can the paper be shortened without overall detriment to the main message?

Yes

Do you think some of the material would be more appropriate as an electronic appendix?

No

Do you have any ethical concerns with this paper?

No

Recommendation?

Major revision is needed (please make suggestions in comments)

Comments to the Author(s)

This article introduces a theoretical connection between the HAVOK approach and the Frenet-Serret fram, which are models constructed from time-delay embedding systems. The authors focuses on the structure of the dynamic matrix forming the HAVOK system and propose an alternative to HAVOK, the sHAVOK that modifies the structure the dynamic matrix: it is diagonal with or without zero elements in HAVOK or sHAVOK, respectively. Depending on the structure of such matrix, the eigenvalues of the dynamical systems change.

The article is well written, and the mathematical equations presented in the article are rigorous and well explained. The authors present a nice review of some elementary concepts of linear algebra, connecting the POD modes (orthogonal basis), with the Grand-Schmid orthogonalization, which is a simple way to build an orthogonal basis. But this is not new. Moreover, there are some major aspects that the authors should address in this article.

- The HAVOK method (1) applies a time-delay to the original data matrix to construct the matrix H , (2) reduces the dimension of matrix H' with a singular value decomposition, as $H' = USV'$, then applies the algorithm DMD with control to matrix V . This is the basis of the algorithm high-order DMD (HODMD). Moreover, HODMD performs an additional dimension reduction in the original data before building the matrix H , very useful in the case of applications with large spatial dimensionality. In the example presented, it is not necessary such dimension reduction, because the authors only analyse signals with 1-4 spatial points, hence it is not possible to reduce the spatial dimension of the system anymore. But it would be interesting that the authors connect both methods discussing major/minor similarities and differences.

- The only difference that I see between the HAVOK and sHAVOK method is that in HAVOK, (1) SVD is applied to the time-delayed snapshot matrix and then (2) the DMD linear relation is built among these snapshots, and in sHAVOK, (1) SVD is applied to the two time-

delayed snapshot matrices that (2) will next conform the DMD linear relation, hence it is not surprising that the matrix A in the second case will be more structured, since the dimension reduction is based on the assumption that the two original snapshot matrices already present a linear relation. Moreover, applying sHAVOK to data with large spatial dimensionality will strongly increase the CPU time and memory, since SVD is applied twice over a large data set. What is the advantage and novelty of sHAVOK? Is it only useful for applications to databases with small spatial dimension?

- Fig. 8: the authors compare the results of (1) HAVOK applied to a small part of the signal, and (2) sHAVOK applied to the same short signal with (3) HAVOK applied to the entire signal, showing that the results of (2) and (3) are more similar, and I guess that suggesting that with sHAVOK it is possible to obtain better results with smaller quantity of data than with HAVOK. I guess that the Lorenz solution that they show is chaotic, and the same in the other two cases. In chaos, if you change the length of the signal, the results will always change. Do the authors choose an arbitrary length for the short signal? Changing this length sHAVOK will also change the results. I want to see if what the authors show is robust, or they are simply selecting a small part of data to obtain what it is interesting. Can you show the reconstruction of the signal when using all the methods? If this is chaos, are you able to reconstruct it?

- Fig. 9: the authors select a very short length of data. What they analyse is still driven by a leading frequency (it is not chaotic), suggesting that the data could be reconstructed when the methodology is properly applied. It is weird that HAVOK is not able to properly identify the dynamics of the system but that sHAVOK is, they are both the same algorithm with the difference in one snapshot in the performance of the SVD. Could this be related to the rank reduction that you perform in the SVD? I would like to see the performance of these method with different ranks. Probably, changing the rank in HAVOK, you will also get a diagonal matrix A , as in sHAVOK. In addition, in DMD with control, what is the forcing frequency that you add? Is it always the same in all the cases? Or do you adapt it to improve the performance of the method in each case? Can you present something that is robust and valid for all the cases?

- How do you choose the length of the short signal in Figs. 8 and 9? Can I see the differences in matrix A as function of the length of this signal?

The theory related to the dynamic matrix A presented in this article is interesting. However, I do not believe that the authors are introducing a new method, and I believe that they should understand first the origin of the differences in matrix A . HAVOK and sHAVOK should provide similar results, since their only difference is a single snapshot in the SVD matrix. To better understand this issue, you could try to see the differences in matrix A using HAVOK and sHAVOK in a periodic system. Please, compare it using different time intervals, different rank reductions for the SVD matrix and, also explain how you choose the forcing frequency. I also would like to see how this matrix A changes as function of the forcing frequency that you choose in the DMD with control. The article is incomplete, and the conclusions are not supported by the results presented. Moreover, the authors need to put an effort on justifying what is the advantage of this algorithm, and explain in detail if it is really working in chaotic systems (as they claim), or clarify that they have only chosen a small portion of a chaotic signal (where there is a leading frequency driving this part of the signal), where they have identified some differences in the matrix A . The application of this methods to chaotic systems should be complemented with more robust results.

Decision letter (RSPA-2021-0097.R0)

10-Jun-2021

Dear Mr Hirsh

The Editor of Proceedings A has now received comments from referees on the above paper and would like you to revise it in accordance with their suggestions which can be found below (not including confidential reports to the Editor).

Please submit a copy of your revised paper within four weeks - if we do not hear from you within this time then it will be assumed that the paper has been withdrawn. In exceptional circumstances, extensions may be possible if agreed with the Editorial Office in advance.

Please note that it is the editorial policy of Proceedings A to offer authors one round of revision in which to address changes requested by referees. If the revisions are not considered satisfactory by the Editor, then the paper will be rejected, and not considered further for publication by the journal. In the event that the author chooses not to address a referee's comments, and no scientific justification is included in their cover letter for this omission, it is at the discretion of the Editor whether to continue considering the manuscript.

To revise your manuscript, log into <http://mc.manuscriptcentral.com/prsa> and enter your Author Centre, where you will find your manuscript title listed under "Manuscripts with Decisions." Under "Actions," click on "Create a Revision." Your manuscript number has been appended to denote a revision.

You will be unable to make your revisions on the originally submitted version of the manuscript. Instead, revise your manuscript and upload a new version through your Author Centre.

When submitting your revised manuscript, you will be able to respond to the comments made by the referee(s) and upload a file "Response to Referees" in Step 1: "View and Respond to Decision Letter". Please use this to document how you have responded to the comments, and the adjustments you have made. In order to expedite the processing of the revised manuscript, please be as specific as possible in your response to the referee(s).

IMPORTANT: Your original files are available to you when you upload your revised manuscript. Please delete any unnecessary previous files before uploading your revised version.

When revising your paper please ensure that it remains under 28 pages long. In addition, any pages over 20 will be subject to a charge (£150 + VAT (where applicable) per page). Your paper has been ESTIMATED to be 27 pages.

Open Access

You are invited to opt for open access, our author pays publishing model. Payment of open access fees will enable your article to be made freely available via the Royal Society website as soon as it is ready for publication. For more information about open access please visit <https://royalsociety.org/journals/authors/open-access/>. The open access fee for this journal is £1700/\$2380/€2040 per article. VAT will be charged where applicable. Please note that if the corresponding author is at an institution that is part of a Read and Publishing deal you are required to select this option. See <https://royalsociety.org/journals/librarians/purchasing/read-and-publish/read-publish-agreements/> for further details.

Once again, thank you for submitting your manuscript to Proc. R. Soc. A and I look forward to receiving your revision. If you have any questions at all, please do not hesitate to get in touch.

Yours sincerely
Raminder Shergill
proceedingsa@royalsociety.org

on behalf of
Dr Bruno Welfert

Board Member
Proceedings A

Reviewer(s)' Comments to Author:

Referee: 1

Comments to the Author(s)

Please see attachment.

Referee: 2

Comments to the Author(s)

This article introduces a theoretical connection between the HAVOK approach and the Frenet-Serret fram, which are models constructed from time-delay embedding systems. The authors focuses on the structure of the dynamic matrix forming the HAVOK system and propose an alternative to HAVOK, the sHAVOK that modifies the structure the dynamic matrix: it is diagonal with or without zero elements in HAVOK or sHAVOK, respectively. Depending on the structure of such matrix, the eigenvalues of the dynamical systems change.

The article is well written, and the mathematical equations presented in the article are rigorous and well explained. The authors present a nice review of some elementary concepts of linear algebra, connecting the POD modes (orthogonal basis), with the Grand-Schmid orthogonalization, which is a simple way to build an orthogonal basis. But this is not new. Moreover, there are some major aspects that the authors should address in this article.

- The HAVOK method (1) applies a time-delay to the original data matrix to construct the matrix H , (2) reduces the dimension of matrix H' with a singular value decomposition, as $H' = USV'$, then applies the algorithm DMD with control to matrix V . This is the basis of the algorithm high-order DMD (HODMD). Moreover, HODMD performs an additional dimension reduction in the original data before building the matrix H , very useful in the case of applications with large spatial dimensionality. In the example presented, it is not necessary such dimension reduction, because the authors only analyse signals with 1-4 spatial points, hence it is not possible to reduce the spatial dimension of the system anymore. But it would be interesting that the authors connect both methods discussing major/minor similarities and differences.

- The only difference that I see between the HAVOK and sHAVOK method is that in HAVOK, (1) SVD is applied to the time-delayed snapshot matrix and then (2) the DMD linear relation is built among these snapshots, and in sHAVOK, (1) SVD is applied to the two time-delayed snapshot matrices that (2) will next conform the DMD linear relation, hence it is not surprising that the matrix A in the second case will be more structured, since the dimension reduction is based on the assumption that the two original snapshot matrices already present a linear relation. Moreover, applying sHAVOK to data with large spatial dimensionality will strongly increase the CPU time and memory, since SVD is applied twice over a large data set. What is the advantage and novelty of sHAVOK? Is it only useful for applications to databases with small spatial dimension?

- Fig. 8: the authors compare the results of (1) HAVOK applied to a small part of the signal, and (2) sHAVOK applied to the same short signal with (3) HAVOK applied to the entire signal, showing that the results of (2) and (3) are more similar, and I guess that suggesting that with sHAVOK it is possible to obtain better results with smaller quantity of data than with HAVOK. I guess that the Lorenz solution that they show is chaotic, and the same in the other two cases. In chaos, if you change the length of the signal, the results will always change. Do the authors choose an arbitrary length for the short signal? Changing this length sHAVOK will also change the results. I want to see if what the authors show is robust, or they are simply selecting a small part of data to obtain what it is interesting. Can you show the reconstruction of the signal when using all the methods? If this is chaos, are you able to reconstruct it?

- Fig. 9: the authors select a very short length of data. What they analyse is still driven by a leading frequency (it is not chaotic), suggesting that the data could be reconstructed when the methodology is properly applied. It is weird that HAVOK is not able to properly identify the dynamics of the system but that sHAVOK it is, they are both the same algorithm with the

difference in one snapshot in the performance of the SVD. Could this be related to the rank reduction that you perform in the SVD? I would like to see the performance of these method with different ranks. Probably, changing the rank in HAVOK, you will also get a diagonal matrix A , as in sHAVOK. In addition, in DMD with control, what is the forcing frequency that you add? Is it always the same in all the cases? Or do you adapt it to improve the performance of the method in each case? Can you present something that is robust and valid for all the cases?

- How do you choose the length of the short signal in Figs. 8 and 9? Can I see the differences in matrix A as function of the length of this signal?

The theory related to the dynamic matrix A presented in this article is interesting. However, I do not believe that the authors are introducing a new method, and I believe that they should understand first the origin of the differences in matrix A . HAVOK and sHAVOK should provide similar results, since their only difference is a single snapshot in the SVD matrix. To better understand this issue, you could try to see the differences in matrix A using HAVOK and sHAVOK in a periodic system. Please, compare it using different time intervals, different rank reductions for the SVD matrix and, also explain how you choose the forcing frequency. I also would like to see how this matrix A changes as function of the forcing frequency that you choose in the DMD with control. The article is incomplete, and the conclusions are not supported by the results presented. Moreover, the authors need to put an effort on justifying what is the advantage of this algorithm, and explain in detail if it is really working in chaotic systems (as they claim), or clarify that they have only chosen a small portion of a chaotic signal (where there is a leading frequency driving this part of the signal), where they have identified some differences in the matrix A . The application of this methods to chaotic systems should be complemented with more robust results.

RSPA-2021-0097.R1 (Revision)

Review form: Referee 1

Is the manuscript an original and important contribution to its field?

Good

Is the paper of sufficient general interest?

Good

Is the overall quality of the paper suitable?

Good

Can the paper be shortened without overall detriment to the main message?

Do you have any ethical concerns with this paper?

No

Recommendation?

Accept as is

Comments to the Author(s)

The authors have adequately addressed the comments of the Referees and in my opinion the paper is publishable at Proceedings A.

Review form: Referee 2

Is the manuscript an original and important contribution to its field?

Good

Is the paper of sufficient general interest?

Good

Is the overall quality of the paper suitable?

Good

Can the paper be shortened without overall detriment to the main message?

Yes

Do you think some of the material would be more appropriate as an electronic appendix?

No

Do you have any ethical concerns with this paper?

No

Recommendation?

Accept as is

Comments to the Author(s)

The authors have responded adequately to all my comments.

Decision letter (RSPA-2021-0097.R1)

09-Sep-2021

Dear Mr Hirsh

I am pleased to inform you that your manuscript entitled "Structured Time-Delay Models for Dynamical Systems with Connections to Frenet-Serret Frame" has been accepted in its final form for publication in Proceedings A.

Our Production Office will be in contact with you in due course. You can expect to receive a proof of your article soon. Please contact the office to let us know if you are likely to be away from e-mail in the near future. If you do not notify us and comments are not received within 5 days of sending the proof, we may publish the paper as it stands.

As a reminder, you have provided the following 'Data accessibility statement' (if applicable). Please remember to make any data sets live prior to publication, and update any links as needed when you receive a proof to check. It is good practice to also add data sets to your reference list. Statement (if applicable): All code used to reproduce results in the figures is openly available at <https://github.com/sethirsh/sHAVOK>.

Citations for data used in manuscript:

K. Kaheman, E. Kaiser, B. Strom, J. N. Kutz, and S. L. Brunton, "Learning discrepancy models from experimental data," in 58th IEEE Conference on Decision and Control. IEEE, 2019

W. P. London and J. A. Yorke, "Recurrent outbreaks of measles, chickenpox and mumps: I. seasonal variation in contact rates," *American journal of epidemiology*, vol. 98, no. 6, pp. 453-468, 1973

Under the terms of our licence to publish you may post the author generated postprint (ie. your accepted version not the final typeset version) of your manuscript at any time and this can be made freely available. Postprints can be deposited on a personal or institutional website, or a recognised server/repository. Please note however, that the reporting of postprints is subject to a media embargo, and that the status the manuscript should be made clear. Upon publication of the definitive version on the publisher's site, full details and a link should be added.

You can cite the article in advance of publication using its DOI. The DOI will take the form: 10.1098/rspa.XXXX.YYYY, where XXXX and YYYY are the last 8 digits of your manuscript number (eg. if your manuscript number is RSPA-2017-1234 the DOI would be 10.1098/rspa.2017.1234).

For tips on promoting your accepted paper see our blog post:
<https://royalsociety.org/blog/2020/07/promoting-your-latest-paper-and-tracking-your-results/>

On behalf of the Editor of Proceedings A, we look forward to your continued contributions to the Journal.

Sincerely,
Raminder Shergill
proceedingsa@royalsociety.org

Reviewer(s)' Comments to Author:

Referee: 2

Comments to the Author(s)

The authors have responded adequately to all my comments.

Referee: 1

Comments to the Author(s)

The authors have adequately addressed the comments of the Referees and in my opinion the paper is publishable at Proceedings A.